# Mitochondrial ATP synthase as a direct molecular target of chromium(III) to ameliorate hyperglycaemia stress

Haibo Wang[1,6], Ligang Hu [1,2,6], Hongyan Li[1,6], Yau-Tsz Lai [1], Xueying Wei[1], Xiaohan Xu[1], Zhenkun Cao[1], Huiming Cao[3], Qianya Wan[4], Yuen-Yan Chang[1], Aimin Xu [5], Qunfang Zhou[2], Guibin Jiang[2], Ming-Liang He[4] & Hongzhe Sun [1] ✉

Chromium(III) is extensively used as a supplement for muscle development and the treatment of diabetes mellitus. However, its mode of action, essentiality, and physiological/pharmacological effects have been a subject of scientific debate for over half a century owing to the failure in identifying the molecular targets of Cr(III). Herein, by integrating fluorescence imaging with a proteomic approach, we visualized the Cr(III) proteome being mainly localized in the mitochondria, and subsequently identified and validated eight Cr(III)-binding proteins, which are predominately associated with ATP synthesis. We show that Cr(III) binds to ATP synthase at its beta subunit via the catalytic residues of Thr213/Glu242 and the nucleotide in the active site. Such a binding suppresses ATP synthase activity, leading to the activation of AMPK, improving glucose metabolism, and rescuing mitochondria from hyperglycaemia-induced fragmentation. The mode of action of Cr(III) in cells also holds true in type II diabetic male mice. Through this study, we resolve the long-standing question of how Cr(III) ameliorates hyperglycaemia stress at the molecular level, opening a new horizon for further exploration of the pharmacological effects of Cr(III).

Chromium is one of the least understood transition metals in the periodic table despite its physiological importance[1–3]. The essentiality, function, and mode of action of Cr(III) in human physiology have been of considerable debate in the field for over half a century[2,4–6]. Since the first report demonstrating that the addition of Cr(III) salts to a rat diet increased blood-glucose removal, Cr(III) has been extensively used as a supplement for the treatment of diabetes mellitus, as well as a nutritional supplement for weight loss and muscle development. Because of the strong implication in maintaining proper carbohydrate and lipid metabolism[7], Cr(III) is the second best-selling mineral supplement

right after calcium in the US[8]. Despite its effectiveness in rodent models, beneficial effects of Cr(III) in humans have not been unequivocally established against Type 2 diabetes mellitus (T2DM)[2,9].

The antidiabetic action of Cr(III) has been well studied in animal models (both db/db insulin-resistance mice and high-fat-induced diabetic mice)[10–12]. It is generally believed that Cr(III) could improve glucose metabolism to maintain normal blood sugar levels, regulate proper carbohydrate and lipid metabolism, and enhance insulin signalling. The effects of Cr(III) on hepatic glycolysis and gluconeogenesis in diabetic mice have been investigated previously in

[1]Department of Chemistry, State Key Laboratory of Synthetic Chemistry, CAS-HKU Joint Laboratory of Metallomics on Health and Environment, The University of Hong Kong, Pok Fu Lam, Hong Kong S.A.R., P.R. China. [2]State Key Laboratory of Environmental Chemistry and Ecotoxicology, Research Center for Eco-Environmental Sciences, Chinese Academy of Sciences, Beijing, P.R. China. [3]Institute of Environment and Health, Jianghan University, Wuhan 430056, P.R. China. [4]Department of Biomedical Science, City University of Hong Kong, Kowloon Tong, Hong Kong, P.R. China. [5]Department of Pharmacology and Pharmacy, and State Key Laboratory of Pharmaceutical Biotechnology, The University of Hong Kong, 21 Sassoon Road, Pok Fu Lam, Hong Kong, P.R. China. [6]These authors contributed equally: Haibo Wang, Ligang Hu, Hongyan Li. ✉e-mail: hsun@hku.hk

different studies[10–12]. It was found that Cr(III) could regulate glyco-lytic enzymes, e.g., glucokinase, phosphofructokinase and pyruvate kinase, and downregulate gluconeogenic enzymes, e.g., glucose-6-phosphatase and phosphoenolpyruvate carboxykinase in the liver of diabetic rats[11,12]. In particular, it has been evidenced that such effects could be attributable to Cr(III)-mediated activation of AMPK signal-ling pathway[13]. Moreover, Cr(III) compounds have also been shown to alter glucose disposal and glucose transport-4 membrane translo-cation (GLUT4) in insulin-resistant mice[10]. Silencing of AMPK abol-ished the protective effects of Cr(III) against GLUT4, resulting in glucose transport dysregulation in skeletal muscle cells[14].

In spite of the extensive studies over several decades, the (bio) chemistry and pharmacology of Cr(III) are still poorly understood. The initially proposed active low molecular weight Cr(III)-binding peptide (chromodulin) has never been reproducibly identified[8]. It is generally believed that Cr(III) could maintain normal blood sugar levels, regulate proper carbohydrate and lipid metabolism, and enhance insulin sig-nalling. However, the underlying molecular mechanism is still unknown[15–17]. The bottleneck of unravelling such mechanisms appears to be the identification of specific biomolecules that account for the physiological effects of Cr(III). These biomolecules have been coined as the "Holy Grail" of chromium biochemistry[18] as they are crucial to confidently verify the physiological or pharmacological effects of Cr(III). Up to now, no Cr(III) direct binding proteins have been unam-biguously identified in cells or tissues apart from the proteins in the blood plasma[19]. The conventional separation and detection methods often result in a poor signal of Cr(III), which might be ascribed to the dissociation of Cr(III) from its bound proteins, and there appears no appropriate approach to track Cr(III)-binding proteins in live cells so far[20].

Herein, we developed a fluorescence-based approach via a Cr(III) probe (Cr$^{3+}$-NTA-AC), successfully visualized the Cr(III)-associated proteome in live cells. These proteins are mainly located in the mito-chondria. We subsequently identified eight Cr(III)-binding proteins and validated them as authentic Cr(III)-binding proteins. We further unravel that mitochondrial ATP synthase serves as an authentic molecular target of Cr(III) to alleviate hyperglycaemia stress both in cells and type II diabetic mice. The role of ATP synthase in the phar-macological effects of Cr(III) is discussed in detail.

## Results

### Mining the Cr(III)-associated proteome in live cells by a fluorescence-based approach

Given that the largest amount of Cr(III) was found in the human liver[21], we thereby attempted to identify Cr(III)-associated proteins in HepG2 liver carcinoma cells treated with Cr$^{3+}$ (as CrCl$_3$) by our home-made continuous-flow gel electrophoresis coupled with ICP-MS (GE-ICP-MS)[22–24], but no Cr$^{3+}$-binding proteins could be found even under non-denaturing conditions (Supplementary Fig. 1a). This could be attributed to the dissociation of Cr$^{3+}$ from its bound pro-teins during conventional separation. To prevent such a dissocia-tion, we therefore sought to develop a strategy to anchor protein targets of Cr$^{3+}$ through a fluorescence probe, i.e., Cr$^{3+}$-bound 2,2′- (5-(2- (7-azido-4-methyl-2-oxo-2H-chromen-3-yl) acetamido)-1-carbox-ylato-pentylazanediyl) diacetate (denoted as Cr$^{3+}$-NTA-AC) (Fig. 1a). The probe Cr$^{3+}$-NTA-AC could label Cr$^{3+}$-binding proteins in live cells and subsequently the labelled proteins could be anchored to the probe through covalent linkage formed upon photo-activation, enabling the separation and identification of these proteins even under denaturing conditions. Given that NTA-AC itself is not cell permeable; only metal-bound NTA-AC (namely M$^{n+}$-NTA-AC or M$^{n+}$-TRACER) can enter cells to label proteins[25]. Previously, different metal-binding proteins have been identified when specific metal ions were bound to NTA-AC, suggesting that the binding of the probe to proteins is metal orientated. Such a strategy has been

successfully employed previously to track different metal-binding proteins in live cells[24,26–29].

The NTA-AC was synthesized by a three-step reaction as reported[25] and the fluorescence probe, Cr$^{3+}$-NTA-AC, was prepared by incubating an equimolar amount of Cr$^{3+}$ (as CrCl$_3$) with NTA-AC in a buffered aqueous solution, and the formation of a 1:1 complex was confirmed by the observation of a peak at m/z 591.1 (calcd. 590.3) in the ESI-MS spectrum (Supplementary Fig. 1b). The response of Cr$^{3+}$-NTA-AC towards proteins was examined by the binding of the probe to bovine serum albumin (BSA) as Cr$^{3+}$ has been previously reported to bind BSA[30]. The binding of Cr$^{3+}$-NTA-AC to BSA resulted in ca. five-fold fluorescence enhancement within 1 h while glutathione (GSH) has negligible effects on the labelling (Supplementary Fig. 1c, d). To visualize the spatial distribution of Cr$^{3+}$ in cells, we treated HepG2 cells with 50 µM Cr$^{3+}$-NTA-AC, where no cytotoxicity was observed (Sup-plementary Fig. 1e), for 30 min at 37 °C. Intense blue fluorescence was observed under the confocal microscope as dots with diameters of ~1 µm in the cytosol of the cells under hyperglycaemia conditions, i.e., 40 mM glucose (Fig. 1b), while tubular shapes appeared when the cells were under low glucose conditions (5.6 mM glucose, Supplementary Fig. 1f). The blue fluorescence signals of Cr$^{3+}$-NTA-AC were partially co-localized with the red fluorescence signals of MitoTracker with colo-calization coefficients of 0.67 ± 0.07 (Fig. 1b), implying that the Cr$^{3+}$-associated proteins (labelled by Cr$^{3+}$-NTA-AC) are mainly localized in the mitochondria of HepG2 cells.

The cells treated with Cr$^{3+}$-NTA-AC were subjected to UV radiation at 365 nm for 10 min prior to lysis to enable the formation of a covalent linkage between the arylazide of the probe and the lit-up proteins. The fluorescently labelled proteins were then separated by two-dimensional gel electrophoresis (2-DE) (Fig. 1c and Supplementary Fig. 2). The protein spots that showed blue fluorescence on the gel image and could be silver-stained simultaneously were excised and subjected to MALDI-TOF mass spectrometry (MALDI-TOF-MS) for protein identification. A total of eight proteins were identified, including two subunits of ATP synthase (ATP5B and ATP5L), two redox-related proteins thioredoxin (TXN) and peroxiredoxin 1 (PRDX1), a heat shock protein mitochondrial Hsp60 (Hsp60), a chloride intra-cellular channel protein 1 (CLIC1) and an enzyme catechol-O-methyltransferase (COMT), among which five proteins were asso-ciated with mitochondria (Fig. 1d and Supplementary Table 1).

To verify that most of the Cr(III)-binding proteins are indeed localized in mitochondria, we examined the cellular uptake and dis-tribution of Cr$^{3+}$ (as CrCl$_3$) in HepG2 cells by ICP-MS. To select the concentration of CrCl$_3$ for the cell study, we first examined the cyto-toxicity of CrCl$_3$ against HepG2 cells and found negligible cytotoxicity of CrCl$_3$ at concentrations lower than 1000 µM (Supplementary Fig. 3a). The cells were therefore treated with 100 µM Cr$^{3+}$ for 8 h at which the maximum cellular uptake was achieved (Supplementary Fig. 3b). The cells were then fractionated into five components (i.e., cytosol, mitochondrion, nucleus, membrane, and others), further digested by nitric acid and subjected to ICP-MS measurement of chromium contents. We found that more Cr$^{3+}$ was uptaken by HepG2 cells under hyperglycaemia condition (1.52 ± 0.05 ng/10$^7$ cells) than that under normal glucose condition (1.17 ± 0.10 ng/10$^7$ cells), and chromium was mainly accumulated in mitochondria (33.3 ± 3.4%), compared to that in cytosol (24.0 ± 2.1%), nucleus (20.1 ± 2.9%), mem-brane (5.9 ± 1.6%) and others (16.7 ± 2.6%) under hyperglycaemia con-dition (Supplementary Fig. 3c). The highest chromium/protein ratio (3.3 µmol/g) was found in mitochondria (Supplementary Fig. 3d), in line with our fluorescence imaging data. In contrast, relatively less accumulation of Cr$^{3+}$ in mitochondria was noted in HepG2 cells incu-bated with the same amount of Cr$^{3+}$ under normal glucose condition (Supplementary Fig. 3e).

To confirm if the identified Cr(III)-associated proteins are authentic Cr(III)-binding proteins, we selected three proteins (ATP5B,

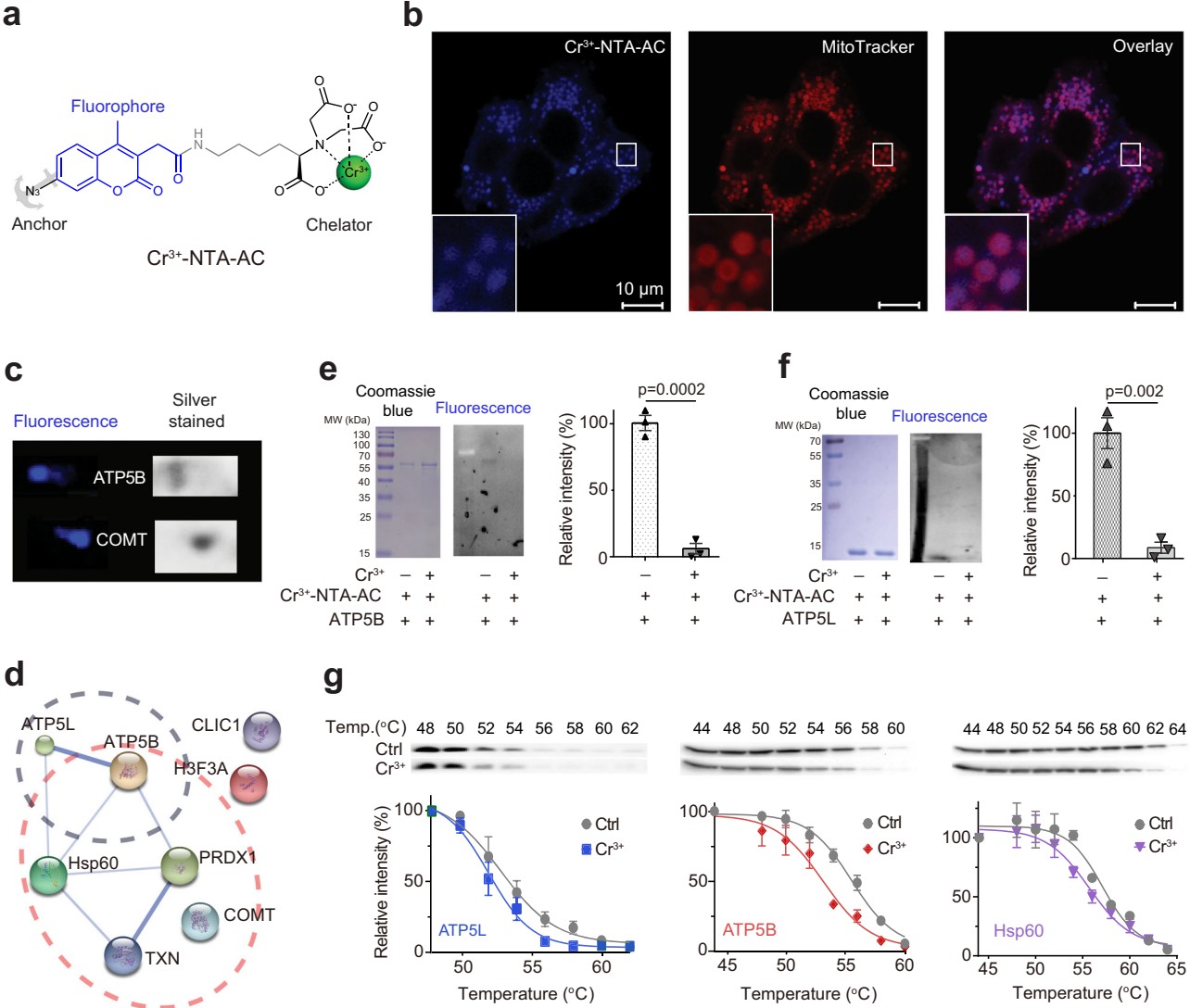

**Fig. 1 | Mining Cr³⁺-associated targets through fluorescent labelling. a** Structure of Cr³⁺-NTA-AC. **b** Representative confocal microscopic imaging of live HepG2 cells under high glucose stress (40 mM) labelled with 50 µM Cr³⁺-NTA-AC and Mito-Tracker red ($n = 5$), demonstrating that most of the blue signals localized in mito-chondria. Scale bar: 10 µm. **c** Selected Cr³⁺-associated proteins in HepG2 cells identified in 2-DE by fluorescence imaging and silver staining. Full 2-DE images are shown in Supplementary Fig. 2. **d** The protein-protein interaction (PPI) network of identified Cr³⁺-associated proteins in HepG2 cells generated via STRING[26]. The PPI network of Cr³⁺-associated proteins was exported from the STRING database (www. string-db.org). **e** Representative SDS-PAGE of ATP5B with Coomassie Blue staining and fluorescence image. $n = 3$; mean ± SEM. Two-sided Student's $t$ test. **f** Representative SDS-PAGE of ATP5L with Coomassie Blue staining and fluores-cence image. $n = 3$; mean ± SEM. Two-sided Student's $t$ test. **g** Cellular thermal-shift assays (CETSA) demonstrating the binding of Cr³⁺ to, ATP5L, ATP5B and Hsp60 in cellulo. $n = 3$; mean ± SEM. One representative result of three independent experi-ments is shown (**b**, **c**). Source data are provided as a Source Data file.

ATP5L and Hsp60) as showcase studies, overexpressed and purified them with their identities being confirmed by MALDI-TOF-MS (Sup-plementary Tables 2-4). To determine the competitive bindings of Cr³⁺ to these proteins with Cr³⁺-NTA-AC, we pre-saturated these proteins with Cr³⁺ (as CrCl₃) prior to the supplementation of Cr³⁺-NTA-AC, and observed over ten-fold higher fluorescence signals for apo-ATP5L, apo-ATP5B or apo-Hsp60 than those for their Cr³⁺ pre-saturated counter-parts on fluorescence images (Fig. 1e, f and Supplementary Fig. 4a), demonstrating that CrCl₃ is able to compete with Cr³⁺-NTA-AC for binding to ATP5L, ATP5B or Hsp60. We further validated the cellular engagement of the identified proteins by Cr³⁺ using a cellular thermal-shift assay (CETSA), a widely used method based on the changes in protein thermal stability upon ligand binding for studies of the target engagement of drug candidates in cells[31]. We treated HepG2 cells with 100 µM Cr³⁺ (as CrCl₃) under hyperglycaemia condition and monitored the thermal changes of the identified Cr³⁺-binding proteins. Treatment

with CrCl₃ resulted in apparent aggregation temperatures (T$_{agg}$, °C) shifted from 52.9 to 52.0, 55.5 to 53.2, 57.2 to 55.9, 65.3 to 61.4, 63.2 to 60.5, 64.1 to 60.8, 56.9 to 51.7, and 59.3 to 53.8 for ATP5L, ATP5B, Hsp60, CLIC1, PRDX1, TXN, COMT, and H3F3A respectively (Fig. 1g, Supplementary Fig. 4b and Supplementary Table 5), confirming that CrCl₃ indeed binds to these identified proteins in cellulo[32]. Similarly, thermal destabilization of these proteins by Cr³⁺-NTA-AC in HepG2 cells was also observed under identical condition with three mito-chondrial proteins Hsp60, ATP5B and ATP5L as showcases (Supple-mentary Fig. 4c), confirming that Cr³⁺-NTA-AC resembles Cr(III) compounds for the identification of protein targets. Nevertheless, the difference in the coordination environment of Cr³⁺-NTA-AC and CrCl₃ could result in a different ligand exchange rate and cell permeability, leading to higher bioavailability of Cr³⁺-NTA-AC over CrCl₃. These results collectively corroborate that the proteins identified by Cr³⁺-NTA-AC are indeed authentic Cr(III)-binding proteins.

## Cr(III) inhibits ATP synthase activity and activates AMPK through binding to the active site of ATP5B

To unveil the vital targets of Cr(III) among the identified proteins, we performed the Gene Ontology (GO) enrichment analysis on cellular component and biological process of these proteins by STRING[33]. The bioinformatics analysis showed that Cr(III)-associated proteins are enriched in mitochondria (Supplementary Fig. 5a) and mitochondrial ATP synthesis (Supplementary Fig. 5b), a process mainly catalyzed by ATP synthase. We therefore hypothesize that the Cr(III)-binding proteins in mitochondria, particularly ATP synthase, may play key roles in mediating the biological effect of Cr(III).

ATP synthase (also known as Mitochondrial Complex V) is responsible for ATP production in the oxidative phosphorylation process and consists of two regions, i.e., the membrane-spanning component ($F_O$) and soluble catalytic core ($F_1$)[34,35]. In the $F_1$ region, three α subunits (ATP5A) and three β subunits (ATP5B) form a hexamer with six ADP/ATP binding sites, among which only the β subunits are capable of catalyzing ATP synthesis/hydrolysis while the α subunits are catalytically inactive[34,35]. As ATP5B has been identified as a $Cr^{3+}$-binding protein both in vitro and in cellulo, we examined the effect of $Cr^{3+}$ on ATP synthase activity in HepG2 cells by ATP Synthase Specific Activity Microplate Assay Kit, in which the activity is measured by ADP production that is coupled to the oxidation of NADH to $NAD^+$, while the enzyme was quantified by a Complex V specific antibody conjugated with alkaline phosphatase. As shown in Fig. 2a, b, the activity of ATP synthase under hyperglycaemia stress was inhibited by $Cr^{3+}$ in a dose- and time-dependent manner with 37.3% activity being inhibited upon supplementation of 200 μM $Cr^{3+}$ to the cells. For comparison, we also treated the cells with an ATP5B inhibitor octyl-α-ketoglutarate (O-KG)[36], and found that 50.8% of the activity was inhibited. A $Cr^{3+}$-mediated inhibition of ATP synthase was observed in C2C12 skeletal muscle cells with slightly higher extent (42.8%) (Supplementary Fig. 6a), demonstrating that such a $Cr^{3+}$-mediated inhibition of ATP synthase could be readily extended to other cell lines. However, supplementation of the same amount of $Cr^{3+}$ to HepG2 and C2C12 cells under normal glucose condition resulted in much less inhibition (11.8% and 15.6%, respectively) of the ATP synthase activity (Supplementary Fig. 6b, c), which might be correlated with less $Cr^{3+}$ uptake as well as less $Cr^{3+}$ accumulation in mitochondria under normal conditions as compared to those under hyperglycaemia conditions (Supplementary Fig. 3e). We next determined AMP/ATP ratios in HepG2 cells upon the treatment of $CrCl_3$ by LC-ESI-MS/MS. The AMP/ATP ratio was significantly increased from $1.8 \pm 0.21$ to $5.6 \pm 0.68$ ($n = 6$) upon the supplementation of $Cr^{3+}$ to HepG2 cells under hyperglycaemia stress (Fig. 2c), in agreement with the ability of $Cr^{3+}$ to inhibit ATP synthase activity, thereby leading to the depletion of ATP production.

AMP-activated protein kinase (AMPK), an energy-status sensor and effector, plays a critical role in maintaining cellular energy homoeostasis[37,38]. When the cell is in energy crisis, AMPK is activated to facilitate the uptake and oxidation of glucose as well as fatty acid, which could be regulated by the cellular AMP/ATP ratio[37,38]. Previously, it has been demonstrated that Cr(III) could activate AMPK to stimulate glucose transport and the silencing of AMPK in skeletal muscle cells (L6-GLUT4myc) abolished the protective effects of $Cr^{3+}$ against glucose transport dysregulation[14]. However, it remains unknown at the molecular level how $Cr^{3+}$ activates AMPK.

We then examined whether $Cr^{3+}$ could also activate AMPK in HepG2 cells under hyperglycaemia condition by measuring AMPK phosphorylation (Thr172) through western blotting. As shown in Fig. 2d, e, the $Cr^{3+}$-mediated time- and dose-dependent increases in AMPK phosphorylation were observed with a ca. two-fold increase compared to the control group, and a similar phenomenon was also observed when treating the cells with the ATP5B inhibitor O-KG. The phosphorylation of acetyl-CoA carboxylase (ACC), a key AMPK target, also increased accordingly (Fig. 2d, e). A similar $Cr^{3+}$-mediated

activation of AMPK and ACC was also observed in C2C12 cells (Supplementary Fig. 7a), which is consistent with a previous study on $Cr^{3+}$-mediated activation of AMPK and ACC in rat L6 muscle cells[14]. In addition, there ware also reports on $Cr^{3+}$-mediated promotion of mitochondrial biogenesis and lipid metabolism in 3T3-L1 adipocytes via the activation of AMPK and ACC[11,39,40]. These observations suggest that $Cr^{3+}$-mediated activation of AMPK under hyperglycaemia condition is not cell or tissue specific. However, supplementation of $Cr^{3+}$ to HepG2 and C2C12 cells cultured in normal glucose condition resulted in no significant phosphorylation of AMPK and ACC (Supplementary Fig. 7b, c). The western blotting results were further confirmed by the AMPK activity assay kit, in which AMPK activity is represented by the contents of P-AMPK with an antibody-based fluorescence approach (Supplementary Fig. 7d, e).

Liver kinase B1 (LKB1) and $Ca^{2+}$/calmodulin-dependent protein kinase kinase-β (CaMKKβ) have been identified as two kinases that phosphorylate Thr172[41]. LKB1 has been found to phosphorylate AMPK by an increased ratio of AMP/ATP in cells[41], while the activation of AMPK by CaMKKβ is initiated by an increase in intracellular $Ca^{2+}$ and does not respond to the change in the ratio of AMP/ATP[42,43]. We thereby further investigated which kinase could be involved in the activation of AMPK. Since the AMP/ATP ratio was significantly increased in HepG2 cells treated with $Cr^{3+}$ under hyperglycaemia stress, we reason that LKB1 might be involved in the activation of AMPK by $Cr^{3+}$. HeLa cells, a LKB1-deficient cell line, was employed to study $Cr^{3+}$-mediated AMPK activation. AICAR, an AMP analogue that activates AMPK through AMP- and LKB1-dependent mechanism and has no effect on AMPKα phosphorylation in HeLa cells[44], was used as the negative control. Different from the observation in HepG2 cells, no obvious increase in AMPK and ACC phosphorylation was observed in HeLa cells treated with increasing concentrations of $CrCl_3$ (Supplementary Fig. 8a), indicating that LKB1 is involved in the activation of AMPK and ACC. We further tested whether CaMKKβ is involved in the activation of AMPK by $Cr^{3+}$. As shown in Supplementary Fig. 8b, c, as compared to the control group, treatment of increasing concentrations of $CrCl_3$ caused almost no change in the level of intracellular $Ca^{2+}$ in HeLa cells, indicating that CaMKKβ is not involved in the activation of AMPK and ACC upon $Cr^{3+}$ treatment. Taken together, we demonstrate that LKB1 but not CaMKKβ is involved in $Cr^{3+}$-mediated activation of AMPK in HepG2 cells.

We then investigated whether the activation of AMPK by $Cr^{3+}$ is attributable to $Cr^{3+}$-induced suppression of ATP synthase activity through targeting ATP5B. The ATP5B in HepG2 cells was knocked down via the transfection of ATP5B siRNA using Lipofectamine RNAi-MAX reagent, while HepG2 cells transfected with scrambled siRNA were used as a negative control. Subsequently, the activity of AMPK in the negative control and ATP5B null HepG2 cells under hyperglycaemia condition was examined in the absence and presence of $CrCl_3$. As shown in Fig. 2f, the expression level of ATP5B in the cells transfected with ATP5B siRNA (10 nM) is ca. 45% of that in the negative control cells. Increases in phosphorylated AMPK ($2.39 \pm 0.11$) and ACC ($1.99 \pm 0.10$) were observed in ATP5B null cells compared to the negative control cells in the absence of $CrCl_3$ (Fig. 2f). However, incubation of the cells with 100 μM $Cr^{3+}$ for 8 h resulted in no significant increase in the levels of phosphorylated AMPK and ACC in ATP5B null cells (Fig. 2f). In contrast, elevated levels of P-AMPK (1.94-folds) and P-ACC (1.71-folds) were observed in the negative control cells after treatment of the same amount of $CrCl_3$ (Fig. 2f), demonstrating that ATP5B is critical for $Cr^{3+}$-mediated activation of AMPK.

We next explored the molecular mechanism underlying $Cr^{3+}$ inhibition on ATP synthase activity. Given that $Cr^{3+}$-ADP/ATP could competitively inhibit mitochondrial $F_1$-ATPase employing $Mg^{2+}$-ADP/ATP as the substrate[45,46], we first examined whether binding of $Cr^{3+}$ induces $Mg^{2+}$ release from ATP5B by ICP-MS. As shown in Fig. 2g, the addition of increasing concentrations of $Cr^{3+}$ and AMP-PNP, a

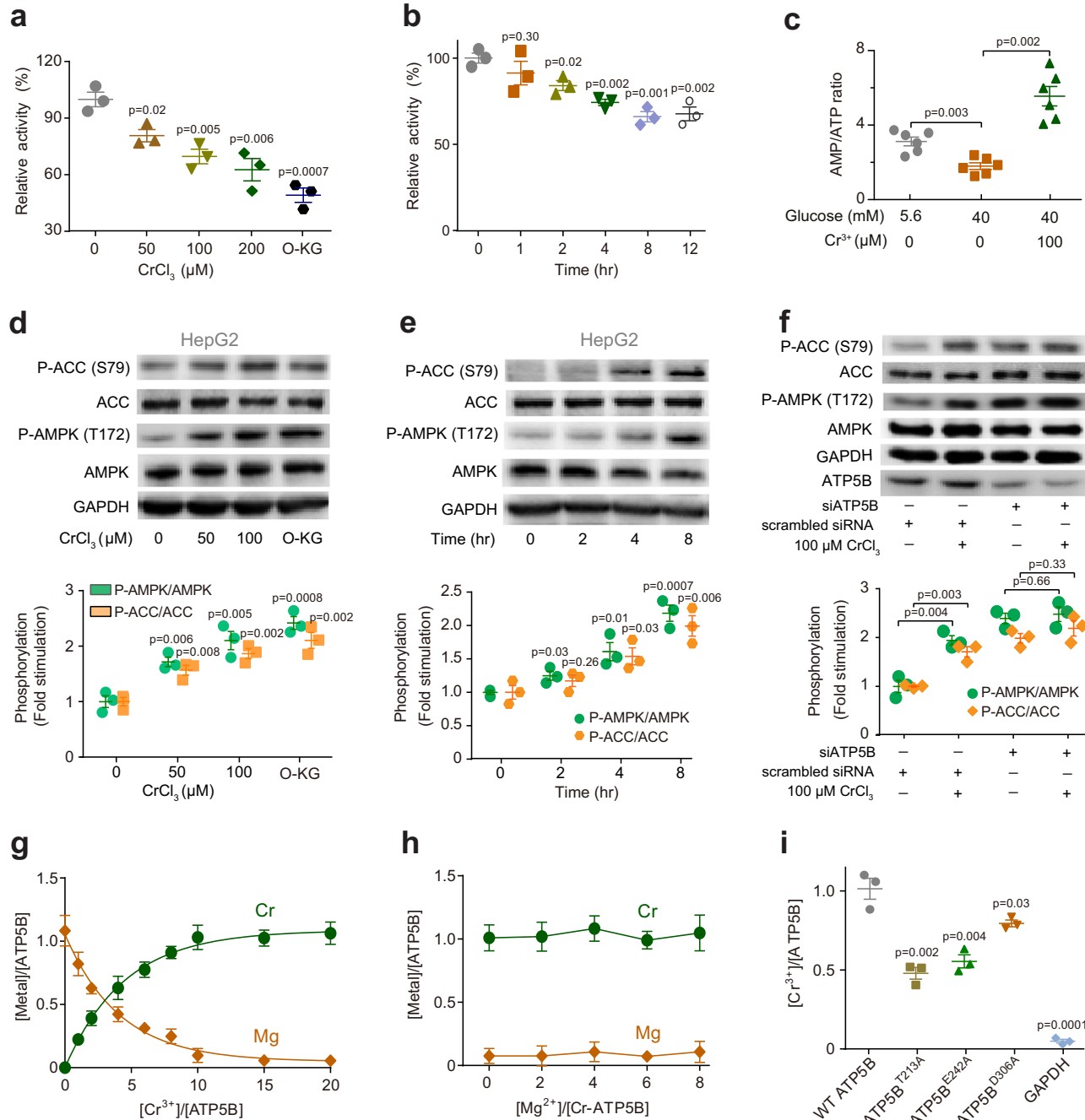

**Fig. 2 | Cr³⁺ attenuates ATP synthase activity and activates AMPK through binding to ATP5B catalytic site. a** Dose-dependent inhibition of ATP synthase activity by Cr³⁺ treatment in HepG2 cells under hyperglycaemia condition. ATP5B inhibitor Octyl-α-ketoglutarate (O-KG) is used as a control. $n = 3$; mean ± SEM. Two-sided Student's $t$ test. **b** Time-dependent inhibition of ATP synthase activity by Cr³⁺ treatment in HepG2 cells under hyperglycaemia condition. $n = 3$; mean ± SEM. Two-sided Student's $t$ test. **c** AMP/ATP ratios in HepG2 cells under different conditions. $n = 6$; mean ± SEM. Two-sided Student's $t$ test. **d** Dose-dependent and (**e**) Time-dependent activation of AMPK and ACC by Cr³⁺ in HepG2 cells under

hyperglycaemia condition. $n = 3$; mean ± SEM. **f** The effect of ATP5B gene silencing on Cr³⁺-mediated AMPK and ACC activation in HepG2 cells. $n = 3$; mean ± SEM. Two-sided Student's $t$ test. **g** The substitution of Mg²⁺ in ATP5B by Cr³⁺ as determined by ICP-MS. $n = 3$; mean ± SEM. **h** The metal contents in Cr-ATP5B upon supplementation of various amounts of Mg²⁺. $n = 3$; mean ± SEM. **i** The Cr³⁺ binding capability of wild-type (WT) ATP5B and various mutants as determined by ICP-MS. $n = 3$; mean ± SEM. Two-sided Student's $t$ test. GAPDH is used as a control. Source data are provided as a Source Data file.

nonhydrolyzable ATP analogue, at a molar ratio of 1:1 into recombinant ATP5B resulted in a gradual decrease in the stoichiometry of Mg²⁺ to ATP5B, accompanied by an increase in the binding ratio of Cr³⁺ to ATP5B. Eventually, ~1.05 molar equivalent (eq.) of Cr³⁺ could completely replace the Mg²⁺ and such a replacement could not be reversed by the addition of excess Mg²⁺ (Fig. 2h), indicating that Cr³⁺ binds to ATP5B at the binding site of Mg²⁺.

To further identify the Cr³⁺ binding residues at the active site of ATP5B, we next mutated the residues that are either directly coordinated to Mg²⁺ (i.e., Thr213) or involved in the hydrogen bonding network with Mg²⁺-coordination ligands (i.e., Glu242 and Asp306) to alanine and examined their Cr³⁺ binding capability. As shown in Fig. 2i, individual site-directed mutation of all these residues resulted in a significant decrease of Cr³⁺ binding to ATP5B, with the highest

decrease of binding stoichiometry being observed for the T213A variant (0.55 eq.), followed by E242A (0.47 eq.) and D306A (0.23 eq.), demonstrating that these residues are indispensable for $Cr^{3+}$ binding. For the GAPDH control group, no detectable $Cr^{3+}$ binding was noted (Fig. 2i). In addition, we found that the absence of nucleotide (e.g., AMP-PNP or ADP) led to a significant drop in $Cr^{3+}$ binding (~ 0.61 eq.) (Supplementary Fig. 6d), indicating that they provide crucial coordination sites for $Cr^{3+}$ binding. Collectively, it is highly possible that the binding of $Cr^{3+}$ to Thr213 and Glu242, which are essential residues for the catalytic process of ATP synthase[47,48], as well as ATP/ADP could lead to the impairment of ATP synthase activity.

It is known that the fully assembled F-type ATP synthase exists as dimers on the inner mitochondrial membrane and this dimerization could affect cristae formation and stabilization[49]. To find out whether $Cr^{3+}$ binding to ATP5B influenced the oligomerization state of ATP synthase, we performed a blue native PAGE (BNP) analysis of digitonin-solubilized mitochondrial proteins. The samples were resolved on a 4%–16% gradient BNP gel capable of resolving proteins at a molecular weight range between 20 to approximately 1, 200 kDa (Supplementary Fig. 9). Silver staining revealed prominent bands at the sizes corresponding to dimeric (~1,000 kDa) and monomeric (~600 kDa) forms of ATP synthase. After treatment with increasing amounts of $Cr^{3+}$, no significant change of the band intensity of either the dimeric or monomeric forms of ATP synthase could be observed, indicating that the oligomerization state of the enzyme was not affected by $Cr^{3+}$ binding.

### Cr(III) rescues mitochondria from hyperglycaemia-induced fragmentation through targeting ATP synthase beta subunit

AMPK activation has been shown to induce manganese superoxide dismutase expression and suppress hyperglycaemia-induced mitochondrial ROS production in endothelial cells, thereby rescuing the mitochondria from fragmentation[50,51]. As $Cr^{3+}$ could activate AMPK through inhibiting ATP synthase by targeting ATP5B, we therefore hypothesize that $Cr^{3+}$ is able to attenuate the hyperglycaemia-induced mitochondrial ROS production and fragmentation.

To test our hypothesis, we first measured the mitochondrial ROS level in HepG2 cells cultured under either normal glucose condition (5.6 mM, as a control) or high glucose condition (40 mM) with or without supplementation of $Cr^{3+}$ by MitoSOX™ Red. As shown in Supplementary Fig. 10a, hyperglycaemia stress led to an increase in the mitochondrial ROS levels of HepG2 cells to 1.5-fold. The treatment of 50 μM CrCl₃ could inhibit the hyperglycaemia-induced mitochondrial ROS by 53.9%, and with the increasing of CrCl₃ concentration, such an inhibition became more evident; in fact, treatment of 100 μM CrCl₃ resulted in almost complete inhibition of hyperglycaemia-induced mitochondrial ROS. For the mitochondrial membrane potential (MMP), hyperglycaemia treatment led to decreased MMP to 55.1% (the MMP under normal glucose condition was set as 100%) and the supplementation of CrCl₃ could reverse the reduced MMP level to normal (Supplementary Fig. 10b). We thereby investigated whether CrCl₃ treatment could attenuate hyperglycaemia-induced mitochondrial fragmentation. HepG2 cells cultured under either normal glucose or high glucose conditions with or without 100 μM $Cr^{3+}$ were stained with MitoTracker and imaged. As shown in Fig. 3a, the mitochondria of HepG2 cells existed predominantly in spherical shape under hyperglycaemia stress, indicative of mitochondrial fragmentation. In contrast, mitochondria remained in their tubular shape post treatment of $Cr^{3+}$, which is similar to those under normal conditions. Quantitative analyses of the mitochondrial morphology based on the values of form factor (FF) and aspect ratio (AR) reveal that supplementation of HepG2 cells with $Cr^{3+}$ rescues mitochondria from hyperglycaemia-induced fragmentation with statistical significance (Mann–Whitney test, $P < 0.001$) (Fig. 3b, c), confirming that $Cr^{3+}$ exerts its effect on glucose metabolism via mitochondria.

To investigate whether the $Cr^{3+}$-mediated rescue of mitochondria is attributed to the targeting of ATP5B, we further examined the morphology of mitochondria in HepG2 cells transfected with scrambled or ATP5B siRNA in the absence or presence of CrCl₃ under hyperglycaemia or normal glucose conditions. Fission and fusion of mitochondria were observed in ATP5B siRNA cells only under hyperglycaemia condition but not under normal glucose condition (Supplementary Fig. 11). Similar to the wild-type HepG2 cells (vide supra), the treatment of 100 μM CrCl₃ could maintain mitochondria in tubular shape in the negative control cells (Supplementary Fig. 11). In contrast, most of the mitochondria remained in spherical shape in the ATP5B-silenced cells (siATP5B) even after treatment of the same amount of CrCl₃ as reflected from the FF and AR (Fig. 3d, e), verifying that silencing of ATP5B renders $Cr^{3+}$ being less effective in rescuing mitochondria from fragmentation. These data collectively demonstrate that $Cr^{3+}$ rescues mitochondria from fragmentation through targeting ATP5B.

### Cr(III) improves glucose metabolism in cells and ameliorates hyperglycaemia stress in diabetic mice

We further investigated whether altered activity of ATP synthase affects the action of $Cr^{3+}$ on glucose metabolism in cells using HepG2 cell under hyperglycaemia stress as an example. As shown in Fig. 4a, CrCl₃ promoted glucose consumption of HepG2 cells by 19.7% (from 23.4 to 28.1 mM/g protein). The silencing of ATP5B also increased the glucose uptake by 23.8% (from 23.4 to 29.0 mM/g protein), while no significant difference in glucose uptake was observed in CrCl₃ treated and non-treated ATP5B null cells, indicating that ATP5B is critical for $Cr^{3+}$-mediated promotion of glucose uptake. In addition, treatment of CrCl₃ resulted in significant upregulation of glycolysis enzymes glucokinase (GCK) ($1.54 \pm 0.06$) and phosphofructokinase (PFKL) ($1.29 \pm 0.03$) but suppression of gluconeogenesis enzymes glucose-6-phosphatase (G6Pase) ($0.86 \pm 0.01$) and phosphoenolpyruvate carboxykinase (PEPCK) ($0.74 \pm 0.02$) (Fig. 4b, c). Silencing of ATP5B resulted in similar effects, i.e., upregulation of GCK ($1.55 \pm 0.03$) and PFKL ($1.32 \pm 0.04$) but suppression of G6Pase ($0.77 \pm 0.06$) and PEPCK ($0.71 \pm 0.04$), compared to cells transfected with scrambled siRNA in the absence of CrCl₃. However, incubation of the ATP5B null cells with 100 μM $Cr^{3+}$ resulted in no significant changes in the levels of GCK, PFKL, PEPCK and G6Pase. These data collectively demonstrate that the promotion of glucose metabolism by $Cr^{3+}$ is attributable to $Cr^{3+}$-induced suppression of ATP synthase activity through targeting ATP5B in cells.

We finally examined the pharmacological effect of $Cr^{3+}$ against hyperglycaemia stress in mice. Herein, both type II diabetic (db/db) and non-diabetic wild-type (WT) mice at 5 weeks old were fed with or without $Cr^{3+}$ (10 mg/kg) through drinking water systems for 11 weeks and subsequently submitted to final tests (Fig. 4d). The treatment of $Cr^{3+}$ led to neither significant alteration of their body weight (less than 10%) nor death of the mice during the whole exposure period (Supplementary Fig. 12a, b). The histologic examination via hematoxylin-eosin-staining on different organ tissues, including kidney, brain, heart, skeletal muscle, and liver, showed no evident changes between the untreated and $Cr^{3+}$-treated mice (Supplementary Fig. 12c). After receiving such a diet for 3 weeks, diabetic symptoms could be observed in the control db/db mice and became more obvious thereafter, whereas significant alleviation of the blood-glucose level was observed in the db/db mice receiving daily oral-dose of $Cr^{3+}$-treatment as compared to those without $Cr^{3+}$-supplementation (Fig. 4e).

We then measured $Cr^{3+}$ contents in different organs of mice and found that the levels of chromium in liver and kidney of $Cr^{3+}$-treated mice were significantly higher than those of the control groups for both db/db and WT mice (Fig. 4f and Supplementary Fig. 13), confirming the bioavailability of $Cr^{3+}$ in vivo. We next examined the ATP synthase activity in mouse livers and found a ca. 25% decrease in the

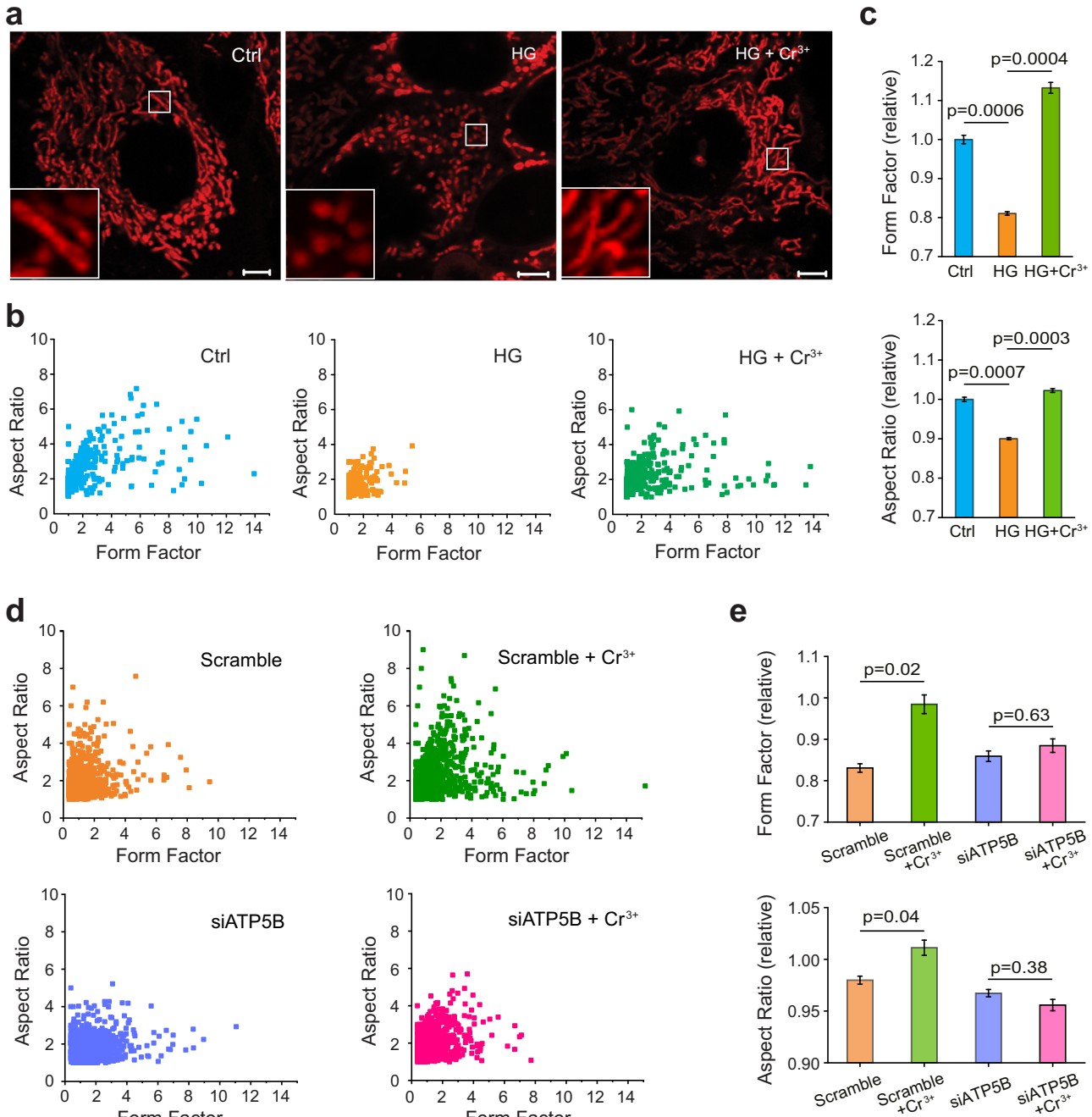

**Fig. 3 | Cr³⁺ rescues mitochondria from hyperglycaemia-induced fragmentation through targeting ATP5B. a** Representative microscopic imaging of HepG2 cells with mitochondria stained by MitoTracker under normal glucose condition (5.6 mM, as Ctrl), high glucose-induced stress (40 mM, as HG) and high glucose-induced stress with Cr³⁺ supplementation (100 μM) (HG + Cr³⁺) (*n* = 10). Inset: 4 times enlargement of mitochondria. Scale bars: 5 μm. **b, c** Quantitative analysis and quantification of mitochondrial morphology illustrated by form factor (FF) and aspect ratio (AR). *n* = 10; mean ± SEM. Two-sided Student's *t* test. **d, e** Quantitative analysis and quantification of mitochondrial morphology in scramble or ATP5B null (siATP5B-transfected) HepG2 cells. *n* = 5; mean ± SEM. Two-sided Student's *t* test. Representative microscopic images are shown in Supplementary Fig. 11. Mitochondria remains fragmented even when 100 μM Cr³⁺ were supplemented to ATP5B-silenced cells. Individual data points (**c, e**) are available from Source Data, which are provided as a Source Data file.

activity of ATP synthase in Cr³⁺-treated db/db mice compared to that of the control group, but not for Cr³⁺-treated WT mice (Fig. 4g). Consistently, an increase in AMP/ATP ratio by 3-folds was observed in the livers of Cr³⁺-treated db/db mice compared to the control group (Fig. 4h and Supplementary Fig. 14a). Moreover, the levels of phosphorylated AMPK and ACC were significantly elevated to 1.83- and 1.48-folds, respectively, in Cr³⁺-treated db/db mice compared to the control group (Fig. 4i), while no significant activation of AMPK and ACC was observed in Cr³⁺-treated WT mice (Supplementary Fig. 14b, c). These

results collectively demonstrate that ATP synthase indeed serves as one of the vital targets of Cr³⁺ in the type II diabetic mouse model (Fig. 4j). We then carried out the oral glucose tolerance test (OGTT) for db/db and WT mice to examine the effect of Cr³⁺ on glucose intolerance. As shown in Supplementary Fig. 14d, a relatively higher glucose elimination rate, as evidenced by a lower glucose level, could be observed in Cr³⁺-treated db/db mice compared to the control group, revealing that Cr³⁺ is able to ameliorate the hyperglycaemia stress in diabetic mice, which is further evidenced by the lower blood glucose

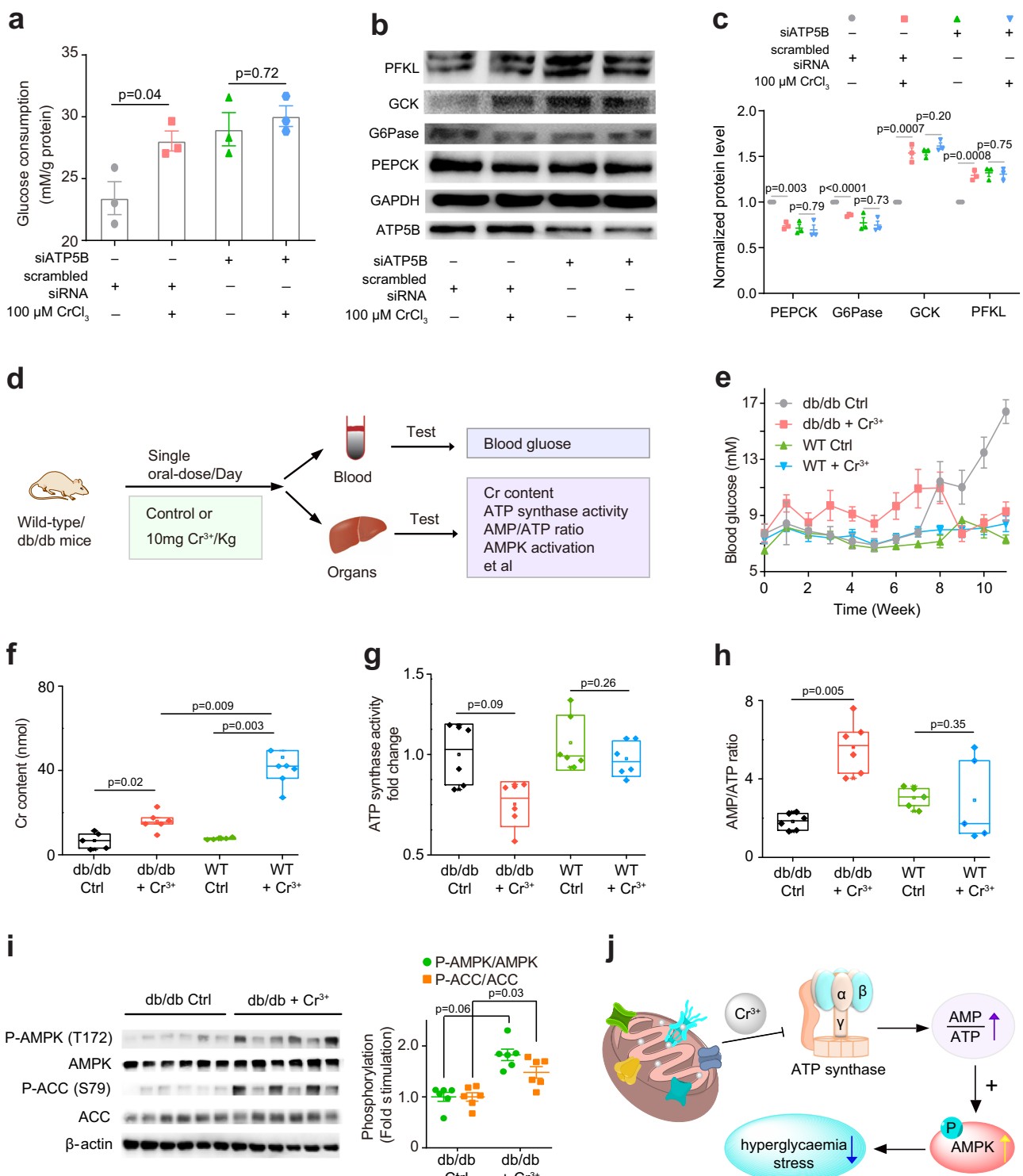

**Fig. 4 | Cr³⁺ ameliorates hyperglycaemia stress through targeting ATP synthase in cells and diabetic mice. a** Comparison of glucose consumption in HepG2 cells. $n = 3$; mean ± SEM. Two-sided Student's $t$ test. Representative western blot (**b**) and quantification results (**c**) showing the effect of ATP5B gene silencing on Cr³⁺-mediated regulation of glycolysis and glycogenesis enzymes in HepG2 cells. $n = 3$; mean ± SEM. Two-sided Student's $t$ test. **d** Schematic illustration of animal study. **e** Blood glucose levels in db/db and wild-type (WT) mice with or without treatment of Cr³⁺ ($n = 6$). **f** Cr³⁺ contents in the livers of both db/db and wild-type (WT) mice with or without treatment of Cr³⁺ ($n = 6$). **g** Cr³⁺ inhibits ATP synthase activity in

db/db mice ($n = 6$). **h** AMP/ATP ratios in the livers of db/db and WT mice with or without Cr³⁺ treatment ($n = 6$). For (**f–h**), centre line, median; box limits, upper and lower boundary, 75 and 25% interquartile ranges respectively; whiskers, maxima and minima; points, all data points. Two-sided Student's $t$ test. **i** Cr³⁺ activates AMPK and ACC in db/db mice. $n = 6$; mean ± SEM. Two-sided Student's $t$ test. **j** Proposed scheme showing that Cr³⁺ ameliorates hyperglycaemia stress through inhibition of ATP synthase and subsequent activation of AMPK in diabetic mice. Source data are provided as a Source Data file.

level in $Cr^{3+}$-treated db/db mice compared to the control group since the eighth week (Fig. 4e). We further examined the effects of $Cr^{3+}$ on hepatic gluconeogenesis in WT and db/db mice. The treatment of $Cr^{3+}$ led to a decrease in the abundance of G6Pase and PEPCK to $0.72 \pm 0.11$ and $0.70 \pm 0.08$ (the control group was set as 1) in the liver of db/db mice, while no significant suppression of G6Pase and PEPCK was observed in $Cr^{3+}$-treated WT mice (Supplementary Fig. 15a, b), demonstrating that $Cr^{3+}$ promotes glucose metabolism and suppresses gluconeogenesis in db/db mice.

## Discussion

Chromium exists in the nature in two common oxidation states, i.e., Cr(III) and Cr(VI), with the trivalent Cr(III) being relatively nontoxic, while hexavalent Cr(VI) compounds are highly redox-active and have long been known as potent toxicants and carcinogens even at low concentrations[52]. Chromate and dichromate are two common forms of Cr(VI) with chromate predominating under physiological conditions. Cr(VI) compounds have been demonstrated to inhibit thioredoxin reductase (TrxR) and oxidize thioredoxin (Trx) and peroxiredoxin (PrX)[52]. There are also controversial reports on the oxidation states of Cr(III) in cells[2,18] and it has been reported that Cr(III) supplements could be readily oxidized to carcinogenic Cr(VI) in living cells[19]. Cr(III) is ubiquitous in food at a very low concentration and has been extensively used as a nutritional supplement. However, the physiological/pharmacological effects of trivalent chromium remain largely obscure and contradictory results exist in both its efficacy and molecular mechanism. The concentrations of chromium are normally in the range of 0.11–16.4 ng/mL and 3–800 ng/g in human blood and different tissues, respectively, in non-exposed populations.

Although proteins/enzymes have long been believed to play crucial roles in the anti-diabetic activity of Cr(III), proteome-wide identification of such targets has not been achieved till now owing to technical challenges. To address this bottleneck, we herein developed a fluorescence-based approach to track Cr(III)-binding proteins via a unique $Cr^{3+}$-NTA-AC probe and identified eight Cr(III)-associated proteins in live HepG2 cells, which are mainly localized in mitochondria. This is in line with an earlier study that $Cr^{3+}$ facilitates tissue-insulin interaction, leading to a considerable increase of mitochondrial swelling[53]. We further verified these proteins as authentic $Cr^{3+}$-binding proteins. These identified $Cr^{3+}$-binding proteins are enriched in the biological process of mitochondrial ATP synthesis, a process mainly catalyzed by ATP synthase. We further demonstrated that Cr(III) binds ATP synthase at its β subunit, the catalytic core of ATP synthase, abolishes its catalytic activity in a dose- and time-dependent manner without significant alteration of its oligomerization state (Fig. 2a, b), leading to the elevation of AMP/ATP ratio (Fig. 2c) and consequently activating AMPK and ACC in HepG2 cells under high glucose stress (Fig. 2f), but not under normal glucose condition. Such a $Cr^{3+}$-mediated inhibition of ATP synthase and activation of AMPK could rescue mitochondria from hyperglycaemia-induced fragmentation (Fig. 3). Finally, we demonstrate that $Cr^{3+}$ ameliorates glucose intolerance in diabetic mice through suppressing ATP synthase activity and activation of AMPK, verifying a proof-of-concept that molecular mechanisms of action of $Cr^{3+}$ in cells could be translated into animal models.

Despite the phenotypic effect of Cr(III) on glucose metabolism has been well established, the underlying molecular mechanism is elusive. In the present study, using HepG2 cells under hyperglycaemia stress as a model, we show that treatment of cells with $Cr^{3+}$ resulted in significant upregulation of glycolysis enzymes (GCK and PFKL) but downregulation of gluconeogenesis enzymes (G6Pase and PEPCK) (Fig. 4b, c). However, silencing of ATP5B abolished such effects, indicating that ATP5B serves as a direct molecular target responsible for $Cr^{3+}$-mediated effects on glucose metabolism.

We further validated the mode of action of Cr(III), i.e., $Cr^{3+}$ inhibits ATP synthase, leading to an elevation of AMP/ATP ratio, and activation

of AMPK, in a mouse model. The db/db mouse model was selected since it has been the most widely used mouse model of T2DM currently[54] and the antidiabetic effects of Cr(III) have also been well investigated in db/db mice[13,55,56]. These studies showed that Cr(III) could effectively decrease the blood glucose level, improve blood lipid metabolism, and increase insulin sensitivity without acute hypoglycaemic side effects in db/db mice. Particularly for the db/db mouse model, it has been reported that Cr(III) activates the AMPK signalling pathway to improve glucose metabolism and suppress gluconeogenesis enzymes of G6Pase and GSK3 to inhibit hepatic glucose production, which consequently ameliorates insulin resistance and attenuates the hyperglycaemic symptom[13]. By using the db/db mouse model, we could better compare our results with the previous findings. Indeed, the results from our study on db/db mice are consistent with previous findings in terms of Cr(III)-mediated activation of AMPK and consequently the decrease of blood glucose. The breakthrough of our current study is that we address the long-standing issue of identifying biomolecules that are responsible for Cr(III) efficacy and unveil the molecular mechanism underlying the antidiabetic effect of Cr(III) (e.g., ameliorate hyperglycaemia and activation of AMPK) through targeting ATP synthase.

Moreover, we investigated the mechanism of $Cr^{3+}$ inhibition of ATP synthase and found that one $Cr^{3+}$ ion binds to the active site of ATP5B with the assistance of the ATP analogue AMP-PNP, leading to the release of one $Mg^{2+}$ cofactor (Fig. 2g). The site-directed mutagenesis study enabled the identification of Cr(III) coordinating residues including Thr213, an $Mg^{2+}$ binding catalytic residue, and Glu242, a non $Mg^{2+}$ coordinated catalytic residue (Fig. 2i). In addition, the nucleotide analogue of AMP-PNP is also involved in $Cr^{3+}$ binding. However, it remains unknown whether other amino acids in the catalytic centre of ATP5B, such as Asp, which is involved in Cr(III)-binding to transferrin[57], also coordinates to or are weakly associated with Cr(III). Thus, the complete coordination geometry of $Cr^{3+}$ to ATP5B warrants further investigation. Nevertheless, we provide direct evidence that $Cr^{3+}$ inhibits ATP synthase via displacement of $Mg^{2+}$ from ATP5B. Previous studies reported that β,γ-Cr(III)ATP and α,β,γ-Cr(III)ATP inhibited $F_1$-ATPase by binding at the catalytic site with the same affinity and α,β-Cr(III)ATP by binding at a regulatory site, and the binding sites showed no significant selectivity for the steric arrangement of the Cr(III) complexes[58,59]. Our findings are reconciling with previous reports that $Cr^{3+}$-ADP/ATP could competitively inhibit mitochondrial $F_1$-ATPase employing $Mg^{2+}$-ADP/ATP as a substrate[45,58–60]. It is worth mentioning that other transition metal ions such as $Zn^{2+}$ and $Mn^{2+}$ could inhibit ATP synthase in vitro[59,61], however, the bindings of these transition metal ions (including $Zn^{2+}$, $Mn^{2+}$, $Cu^{2+}$, $Co^{2+}$, $Ni^{2+}$, and $Fe^{3+}$) are not as strong as $Cr^{3+}$ as evidenced by the binding affinities and competition studies (Supplementary Fig. 16a, b). Thus, they are unlikely to compete with $Cr^{3+}$ towards binding to ATP synthase, particularly in cells owing to the low cellular concentration of free transition metal ions (i.e., tightly regulated).

AMPK is a well-known energy sensor in human cells, which could be activated when cells are in energy crisis and has been widely employed as a therapeutic target for type 2 diabetes[62,63]. Activation of AMPK stimulates glucose uptake and fatty acid oxidation in skeletal muscle and heart but inhibits fatty acid synthesis in liver and lipogenesis in adipose tissue[64]. Despite Cr(III) has been shown to activate AMPK to stimulate glucose transport, and silencing of AMPK abolished the protective effects of Cr(III) against glucose transport dysregulation[14], the underlying molecular mechanism of Cr(III) induced AMPK activation remains elusive. Our studies provide the molecular evidence on the mechanism of Cr(III)-mediated AMPK activation, i.e., $Cr^{3+}$ activates AMPK indirectly through targeting ATP synthase (mitochondrial complex V) beta subunit. Such indirect activation of AMPK was also observed for the first-line anti-diabetic drug, metformin, which inhibits mitochondrial complex I to alter cellular

energy metabolism[65]. Although complete loss of mitochondrial function is detrimental and lower hepatic level of ATP has been linked to type II diabetes mellitus[66], partial suppression of the electron transport chain has been shown to be beneficial to mammals[67]. Recently, it has been demonstrated that a metabolite, α-ketoglutarate, is able to extend the lifespan in *Caenorhabditis elegans*, attributable to the inhibition of ATP synthase activity[36].

Mitochondria are highly dynamic organelles that undergo coordinated cycles of fission and fusion (referred as mitochondrial dynamics) to maintain their morphological plasticity and physiological functions[68]. More fission results in fragmented mitochondria while excess fusion leads to elongated mitochondrial tubules. Increasing evidence suggests that mitochondrial dynamics play vital roles in metabolism-secretion coupling in pancreatic β-cells and complications of diabetes[69]. High glucose concentration could induce mitochondrial fragmentation in various cell lines, which is linked to the hyperglycaemia-induced mitochondrial ROS production[51,69]. AMPK activation has been proved to normalize hyperglycaemia-induced mitochondrial ROS production and rescue mitochondria from fragmentation[50,51]. Our results showed that CrCl₃ treatment reversed the hyperglycaemia-induced mitochondrial ROS increase and MMP decrease. Further imaging study revealed that the supplementation of Cr³⁺ attenuated mitochondria morphology alteration from a tubular shape to a spherical shape induced by high glucose stress. Such a Cr³⁺-mediated shift of fission-fusion balance rescued mitochondria from hyperglycaemia-induced fragmentation. Further gene silencing experiment evidenced that the rescue of mitochondria by Cr³⁺ is attributable to its targeting of ATP5B and consequent inhibition of ATP synthase and activation of AMPK.

In summary, we developed a robust approach by integrating fluorescence imaging with high throughput proteomic analysis, enabling the endogenous Cr(III) associated proteome to be identified in live cells. We further validated ATP synthase as an authentic molecular target to ameliorate hyperglycaemia stress. Through this study, we resolve the long-standing issue on how Cr(III) ameliorates hyperglycaemia stress at the molecular level. Given that multiple Cr(III)-binding proteins have been identified, this study opens a new horizon for further investigation of pharmacological aspects of Cr(III) other than the anti-diabetes, such as anti-neurodegeneration and anti-aging.

## Methods

### Materials

The study complies with the ethical regulations with all experiment protocols were approved by Committee of Scientific Research of the University of Hong Kong. Animal experiments were performed in accordance with the Animal Use Protocol approved by Committee of Scientific Research in Research Centre for Eco-Environmental Sciences, Chinese Academy of Sciences. HepG2 (Catalogue No. HB-8065), Hela (Catalogue No. CRM-CCL-2) and C2C12 myoblast cell line (Catalogue No. CRL-1772) were purchased from American Type Culture Collection (ATCC). Dulbecco's Modified Eagle's Medium (DMEM), fetal bovine serum (FBS), horse bovine serum, MitoTracker® Red CMXRos, ATP5B siRNA, negative control siRNA, lipofectamine RNAiMAX Reagent and OPTI MEM medium were purchased from Invitrogen (Carlsbad, CA). The ATP5B-specific siRNA (Catalogue No.: 4390824) and negative control siRNA (Catalogue No.: 4404021) were obtained from Thermo Scientific. Chromium chloride (CrCl₃), ICP-MS multi-element standard solution (51844), cell proliferation kit II (XTT), and all other chemicals were obtained from Sigma-Aldrich (St. Louis, MO. USA). ATP Synthase Specific Activity Microplate Assay Kit, antibodies against ATP5B, ATP5L, Hsp60, β-actin, phospho-AMPK-α (Thr172), AMPK, phosphoacetyl CoA carboxylase (ACC, Ser 79) ACC and GAPDH were from Abcam (Cambridge, UK). Antibodies against CLIC1, PRDX1, TXN, COMT, H3F3A, PEPCK, G6Pase, GCK and PFKL were purchased from Abclonal (China). EnzyFluo™ AMPK Phosphorylation Assay Kit was purchased from BioAssay Systems (Hayward, USA). Bovine serum

albumin (BSA) was purchased from USB Chemicals and used as received. All fluorescence measurements were performed on a Hitachi F-7000 fluorescence spectrophotometer with 1000 W xenon lamp using a 1 cm × 1 cm quartz cuvette (1.5 mL sample volume).

### (S)-2,2′-(5-(2-(7-azido-4-methyl-2-oxo-2H-chromen-3-yl)))diacetate synthesis

The fluorescent ligand (S)-2,2′-(5-(2-(7-azido-4-methyl-2-oxo-2H-chromen-3-yl)))diacetate (i.e., NTA-AC) was synthesized and characterized in our previous report[25]. In detail, 7-amino-4-methylcoumarin-3-acetic acid (0.143 g, 0.61 mmol) was first mixed with sodium nitrite (0.045 g, 0.65 mmol) in water (6 mL) with concentrated sulphuric acid (1 mL) in ice bath. Sodium azide (0.051 g, 0.79 mmol) was then added, and the reaction mixture was continuously stirred for 45 min for precipitate to be formed. The precipitate was subsequently filtered, washed with ice-cold water, and dried by lyophilization to obtain a light brown powder as 2-(7-azido-4-methyl-2-oxo-2H-chromen-3-yl) acetic acid (0.135 g, 0.52 mmol, 85% yield). The brown powder (0.129 g, 0.50 mmol) was then dissolved in 25 mL of acetonitrile with stirring at room temperature. N-hydroxysuccinimide (0.058 g, 0.51 mmol) and N,N′-dicyclohexylcarbodiimide (0.105 g, 0.51 mmol) were subsequently added to the reaction mixture and the reaction mixture was stirred overnight at room temperature. Afterwards, the solution was filtered, and the filtrate was rotary-evaporated to yield a crude yellow product, which was then dissolved in chloroform for simple solvent extraction. After rotary-evaporation, crude yellow solid as 2,5-dioxopyrrolidin-1-yl 2-(7-azido-4-methyl-2-oxo-2H-chromen-3-yl) acetate was obtained. The crude yellow solid was dissolved in acetonitrile (50 mL) with stirring at room temperature. N$_α$,N$_α$-bis(carboxymethyl)-L-lysine hydrate (0.180 g, 0.69 mmol) was dissolved in water (10 mL) supplemented with a small volume of triethylamine (0.5 mL). The solution was subsequently added dropwise to the crude yellow solid solution, and the reaction mixture was stirred overnight continuously at room temperature. After rotary-evaporation and lyophilization, the crude product was purified through column chromatography to obtain pale yellow powder of NTA-AC. The NMR characterization of NTA-AC could be found in previous report[25].

### Probe fluorescence spectroscopic measurements

The probe Cr³⁺-NTA-AC was generated by reacting Cr³⁺ (as CrCl₃) with equimolar amount of fluorescent ligand (20 mM Tris-HCl, pH 7.2) and the formation of 1:1 complex was confirmed by ESI-MS. Fluorescence responses of Cr³⁺-NTA-AC towards proteins were determined by reacting 5 μM Cr³⁺-NTA-AC with different molar equivalents of proteins at 25 °C for 30 min. The fluorescence spectra were then recorded, and the fluorescence was quantified.

### Mammalian cell culture

HepG2 cells were grown in DMEM supplemented with 10% fetal bovine serum (FBS) and 1% antibiotics (Pen Strep), and cells were all cultured in 5% CO₂ incubator at 37 °C. Glucose was added to 40 mM for cells cultured under high glucose condition. Hank's Balanced Salt Solution (HBSS) was used to replace culture medium for probe-related experiments prior to imaging.

Mouse C2C12 myoblast cells were maintained in DMEM supplemented with 10% FBS and 1% antibiotics (Pen Strep) in a humidified atmosphere of 95% air and 5% CO₂ at 37 °C. At the confluency of 90%, the myoblasts cells were then rinsed by PBS and switched to DMEM containing 2% horse serum for differentiation. Fresh differentiation medium was fed to the cells every 24 h and lasted for 5 days until over 90% of the myoblast cells were differentiated to myotubes. The C2C12 myotubes were used for further experiments.

### Examination of the cytotoxicity of Cr³⁺-NTA-AC and CrCl₃

HepG2 cells were seeded into 96-well plates (10, 000 cells/well) and incubated with respective medium overnight at 37 °C with 5% CO₂. The

medium was replaced, and the cells were incubated with different concentrations (0, 25, 50 and 100 μM) of Cr$^{3+}$-NTA-AC in medium for 30 min at 37 °C and protected from light. For CrCl$_3$, different concentrations (0, 50, 100, 200, 400, 600, 800 and 1000 μM) of the compound were added in medium for 24 h at 37 °C. The medium was then removed and replaced with fresh medium, 10 μL of (3-(4,5-dimethylthiazol-2-yl)-2,5-diphenyl-tetrazolium bromide (MTT, 5 mg/mL in sterile PBS)) were added and further incubated for 4 h. After removing all but 25 μL of medium, 50 μL of DMSO were applied to incubate for 10 min at 37 °C. The absorbance of each well was recorded at 540 nm using a microplate reader (BIO-RAD, iMark™), and cell viability is reported relative to those of untreated cells.

### Fluorescence imaging of HepG2 cells
HepG2 cells were transferred onto a confocal dish for confocal imaging and allowed to stably adhere to the glass bottom for 2 days. Afterwards, DMEM was removed from the culturing plate and cells were washed with 37 °C HBSS once. Solution was then removed and replaced by HBSS preloaded with 50 μM Cr$^{3+}$-NTA-AC and 50 nM MitoTracker® Red CMXRos (Molecular Probes) to further incubate for 30 min. The buffer solution was then discarded, and cells were washed by HBSS for three times and subjected to confocal imaging. Fluorescent and phase contrast images were captured by a Carl Zeiss LSM700 Inverted Confocal Microscope with the use of 405 nm and 555 nm lasers under a Plan-Apochromat 63×1.40NA oil-immersion objective, while the ranges of emission were fixed at $\lambda_{em}$ = 430–500 nm (for Cr$^{3+}$-NTA-AC) and 590–650 nm (for MitoTracker Red) for imaging.

### Identification of Cr(III)-binding proteins
After incubating HepG2 cells with 50 μM Cr$^{3+}$-NTA-AC for 30 min, the cells were washed three times with HBSS and ultraviolet irradiated at 365 nm for 10 min to allow complete photoactivation of the arylazide. Cr$^{3+}$-NTA-AC-labelled/unlabelled samples were then washed twice with ice-cold PBS (pH 7.4) and trypsinized to collect the cells via centrifugation. Cells were then resuspended in 20 mM HEPES, 100 mM NaCl, pH 7.2 with the supplementation of protease inhibitor cocktail (Sigma-Aldrich) and 1 mM TCEP, and subsequently lysed by sonication to separate supernatant (cytosolic fraction) and insoluble debris (fraction including membrane and unlysed organelles). Insoluble fractions of probe-labelled/unlabelled samples were washed three times by buffer and dissolved in PBS buffer at pH 7.4 supplemented with 1% SDS. Separation of proteins in different fractions was performed by 2-DE or SDS-PAGE. Protein spots and bands corresponding to the fluorescent ones were excised, trypsin digested and identified by Matrix Assisted Laser Ionization Time-of-Flight Mass Spectrometry (MALDI-TOF-MS).

### Bioinformatics analysis
The protein-protein interaction network of Cr(III)-associated proteins was obtained and visualized as confidence view via STRING 9.1 (www.string-db.org) with a confidence score of 0.4. Further enrichment analysis on the identified Cr(III)-associated proteins was also performed on the platform[33]. The GO term with a p-value (False Discovery Rate corrected) less than 0.05 was considered as statistically enriched.

### Cellular thermal shift assay
In order to validate the Cr(III)-binding proteins identified by Cr$^{3+}$-NTA-AC, cellular thermal shift assay was performed according to a previous report[70]. In detail, HepG2 cells were treated with CrCl$_3$ (100 μM) or Cr$^{3+}$-NTA-AC (100 μM) for 16 h, harvested, and washed with PBS for three times. Cells were resuspended in ice-cold PBS. Afterwards, 100 μL of cell suspension was transferred into 0.2 mL PCR tubes, and each tube was heated in parallel in a PCR machine for 6 min to the respective temperature (44 to 84 °C). The tubes were then incubated

for 6 mins at room temperature. Subsequently, samples were frozen in liquid nitrogen for 3 min and then thawed in a 25 °C water bath for 3 min. This freeze-thaw cycle was repeated five times. The entire content was centrifuged at 16,000 × g for 20 min at 4 °C. After centrifugation, 70 μL of each supernatant was transferred into a new tube. The protein concentration of the 44 °C sample was determined and used to normalize the volume of samples in SDS-PAGE. Proteins in the supernatant were denatured using the SDS sample buffer, partially separated by means of SDS gel electrophoresis, and subjected to western blot analysis using anti-ATP5B, anti-ATP5L, anti-CLIC1, anti-PRDX1, anti-TXN, anti-COMT, anti-H3F3A, anti-GAPDH and anti-Hsp60 antibodies. The band intensities were quantified by Image J (version 1.52d) and analyzed with Boltzmann sigmoidal curve fit in Graphpad Prism software (version 6.01). Results were represented as mean ± SEM of three independent experiments.

### Cellular uptake and distribution of chromium
HepG2 cells were seeded into 6-well plates and treated with 100 μM of Cr$^{3+}$ at the confluency of 80%. Cells were harvested after Cr$^{3+}$-treatment for 1, 2, 4, 6, 8, 12, 18 h. Cell pellets with same amounts of cell numbers (1 × 10$^6$) were collected and digested with 100 μL 30.0% HNO$_3$ (Fluka, 84385) overnight, which was diluted to a final volume of 1 mL with 1% HNO$_3$. Samples were further diluted when the measured $^{52}$Cr signals exceeded the linear range of the standard curve. The chromium contents in HepG2 cells were calculated according to the standard curve and normalized to the number of atoms in single cells.

To measure the distribution of Cr$^{3+}$ in different cellular components, HepG2 cells were harvested after treatment of 100 μM Cr$^{3+}$ (as CrCl$_3$) for 8 h. The cells were washed with ice-cold Tris-HCl buffer (100 mM, pH 7.4) for three times. The cell pellets were re-suspended in 5 packed cell pellet volumes of hypotonic buffer A (10 mM HEPES, 10 mM KCl, 1.5 mM MgCl$_2$, pH 7.5) on ice for 10 min. Subsequently, the cells were lysed by 10 strokes of a Kontes all-glass Dounce homogenizer that was washed with ultrapure water and then rinsed with buffer A. The homogenate was checked microscopically for cell lysis and centrifuged for 10 min at 3000 × g to get the nuclei pellet and supernatant I. The supernatant I was centrifuged at 20, 000 × g for 30 min to get the rude mitochondria pellet and supernatant II. The supernatant II was further fractionized by centrifugation at 100,000 × g to get the membrane protein (the pellet) and cytosolic protein (the supernatant III). The pellets obtained in each step were washed with buffer A for 3 times and then resuspended in buffer B (20 mM HEPES pH 7.5, 420 mM NaCl, 1.5 mM MgCl$_2$, 25% glycerol, 1% NP-40), following by sonication (amplitude: 20%, 5 s on, 10 s off, in total 1 min) on ice-water bath for lysis. Protein concentrations were determined by bicinchoninic acid (BCA) assays. The protein samples were further digested with 50% HNO$_3$ overnight and diluted with 1% HNO$_3$ before submitting for ICP-MS measurement.

### ATP synthase activity
HepG2 cells and C2C12 myotubes were subjected to high glucose stress to evaluate the effects of Cr$^{3+}$ on the activities of ATP synthase. ATP synthase activity was assayed with ATP synthase specific activity microplate assay kit (ab109716, Abcam) according to the manufacturer's protocol, in which the activity was measured by ADP production that is coupled to the oxidation of NADH to NAD$^+$; while the quantity of this enzyme was determined by Complex V specific antibody conjugated with alkaline phosphatase. Briefly, HepG2 and C2C12 muscle cells were cultured either under normal condition as a control (5.6 mM D-glucose) or high level of D-glucose (40 mM) in 6-well plates, while CrCl$_3$ was added to the culture medium with the final concentrations of 0, 20, 50, 100 and 200 μM. ATP5B inhibitor Octyl-α-ketoglutarate (O-KG) (400 μM) was used as control. After incubation for 8 h, the cells were collected and resuspended in Tris-HCl buffer (100 mM, pH 7.4, 25% glycerol, 1× protease inhibitor cocktail). The

samples were then lysed through sonication (amplitude 20%, 5 secs on, 20 secs off, in total 30 sec on the ice-water bath), followed by centrifugation at 16, 000 × g and the supernatants were then collected. An aliquot of solution (typically with ca. 50 µg proteins in 50 µL solutions) was applied to the microplate assay kit and incubated at 4 °C overnight. The ATP synthase was captured by the immobilized monoclonal antibody in the wells. After a wash step, the development agents were added, and the absorbance was measured at the interval of 1 min for 60 min at 37 °C. The ATP synthase activity is expressed as the rate of absorbance changes. Afterwards, the captured ATP synthase in each well was quantified with a secondary antibody, and the final enzyme activities were presented as normalized value for comparison.

### Protein expression and purification
The plasmids for wild-type ATP5B and ATP5B mutants were constructed into pET30a vector by GenScript (Hong Kong). The plasmids were transformed into BL21 (DE3) cells for protein expression. In brief, overnight cultures of BL21 (DE3) cells harbouring corresponding plasmid were diluted by 1:100 to fresh Luria-Bertani medium supplemented with 50 µg/mL kanamycin. Cells were grown at 37 °C with rotation of 200 rpm until the $OD_{600}$ reached 0.6. ATP5B expression was induced by addition of 500 µM isopropyl-β-D-thiogalactoside (IPTG) and the bacteria were further cultured for 12 h at 37 °C. The bacteria were harvested by centrifugation (5000 × g, 20 min at 4 °C) and the cell pellets were resuspended with 25 mM Tris-HCl (pH = 7.4), 100 mM NaCl buffer and lysed by sonication. The lysates were centrifuged at 1, 5000 × g for 30 min, and the supernatant was collected and then applied to a 5 mL HisTrap Q column (GE Healthcare). The proteins were eluted with 300 mM imidazole in 25 mM Tris-HCl (pH = 7.4), 100 mM NaCl. They were further purified by a HiLoad 16/60 Superdex 200 column equilibrated with Tris-HCl buffer. The identity of the purified ATP5B was further confirmed by MALDI-TOF MS.

### Metal content determination in purified proteins
The metal contents in proteins were determined by ICP-MS (Agilent 7700x system). In detail, wild-type ATP5B and ATP5B mutants were incubated with different amount of metal ions (including $CrCl_3$, $CoCl_2$, $NiCl_2$, $CuCl_2$, $ZnCl_2$, $MnCl_2$, and $FeCl_3$) and AMP-PNP at the molar ratio of 1:1 in Tris-HCl buffer (pH 7.4). After incubation at room temperature for 2 h, excess amounts of metal ions were removed by dialysis. The protein concentrations were determined by a bicinchoninic acid (BCA) assay. The standard curves of different metal ions (0, 1, 5, 10, 50 and 100 ppb) were prepared from a multielement standard solution (90243, Sigma-Aldrich) for ICP-MS. Metal contents were measured in triplicates and calculated based on the standard curve. For the competition study, 10 eq. of different metal ions were added into Cr(III)-ATP5B and incubated overnight. The excess metal ions were removed by dialysis and samples were subjected to ICP-MS measurement.

### ATP synthase extraction and blue native page analysis
Hela cells were harvested and washed with chilled PBS (pH 7.4) for three times. The harvested cell pellets were re-suspended in hypotonic buffer A (10 mM HEPES, 10 mM KCl, 1.5 mM $MgCl_2$, pH 7.5) with the presence of protease inhibitor and then lysed by Vibra-Cell Ultrasonic Processor with Sound Abating Enclosure. The cell debris and unbroken cells were removed by centrifugation (3000 × g for 10 min at 4 °C). The supernatant was further centrifuged (20,000 × g for 30 min at 4 °C) and mitochondria were collected from the pellet. The mitochondria were lysed by 1% digitonin in ice-water bath for 30 min. The solubilized mitochondria lysate was collected by centrifugation of 30 min at 20,000 × g. The supernatants were treated with benzonase and centrifuged again. The soluble complexes were analyzed by blue native page (BNP) 4 °C in 4–16% acrylamide gradient Bis-Tris gels (Thermo Fisher Scientific) according to the manufacturer's instructions.

### Determination of AMP/ATP ratio
To evaluate the effects of $Cr^{3+}$ on the AMP/ATP ratio, HepG2 cells were subjected to high glucose stress, followed by administration of $CrCl_3$. The AMP and ATP in various HepG2 samples were extracted and measured with LC-ESI-MS/MS. Briefly, HepG2 cells were cultured either under normal condition as a control (5.6 mM D-glucose) or high level of D-glucose (40 mM) in 100 mm culture dishes supplemented with $CrCl_3$ at a final concentration of 100 µM to the culture medium. After incubation, the cells were added to 300 µL extraction buffer (10 mM ammonium acetate with pH 7.5, 4 mM EDTA, saturated with phenol), scraped off, and lysed via sonication. The samples were then centrifuged at 13,500 × g, and the supernatants were collected. Upon filtering by 0.22 µm membrane, the cell lysates were subjected to HPLC-ESI-MS/MS (Waters) for detection of AMP and ATP immediately. HPLC condition: 5% A (10 mM ammonia acetate)/95% B (methanol with 0.1% formic acid) for 0–5 min, followed by gradual increase to 100% A from 5–16 min, then quickly changed to 5% A/95% B and kept till the end (16–21 min). Adenosines were measured by ESI-MS in a negative mode. The analytes were monitored in a multiple reaction monitoring (MRM) mode with ion transitions of 506.6 to 159.2 for ATP and 345.7 to 79.0 for AMP respectively. The detailed mass spectrometry parameters for each ion transitions were: for ion transition of 506.6 to 159.2 (ATP), cone voltage and collision voltage were 30 v and 36 v; for ion transition of 345.7 to 79.0 (AMP), cone voltage and collision voltage were 30 v and 30 v. Other parameters include source temperature of 120 °C; desolvation temperature of 450 °C; desolvation gas flow of 800 L/h and cone gas flow of 50 L/min. The contents of AMP and ATP in the samples were determined based on a calibration curve.

### ATP5B gene knockdown
HepG2 cells were seeded in 12-well culture plates with a density of $2 × 10^5$. Cells were further cultured overnight and then transfected with siRNA using lipofectamine RNAiMAX Reagent (Invitrogen). In brief, 10 pmol of siRNA was diluted into 50 µL Opti-MEM media, which was mixed with 15 µL lipofectamine RNAiMAX Reagent (dilluted to 50 µL with Opti-MEM medium). The mixture was further incubated for 20 min at room temprature and subsequently applied to the cell cultures. Cells were harvested and lysed after transfection for 24 and 48 h for silencing condition optimization. The abudance of ATP5B was examined and quantified via western blot.

### Western blotting analysis
Western blot analysis was performed according to the standard protocol. In detail, HepG2 or Hela cells after treatment of various conditions were harvested and lysed through cell lysis buffer. Cell debris was removed by centrifugation (16,000 × g for 20 min at 4 °C). The supernatant was collected, and the protein concentration was determined by BCA assay. Generally, 30 µg of proteins were separated by polyacrylamide gel electrophoreses (SDS-PAGE). After SDS-PAGE separation, proteins of interest were electrophoretically transferred to polyvinylidene difluoride (PVDF) in a transfer buffer (20 mM Tris-HCl, 154 mM glycine and 20% methanol). The membranes were blocked with 5% (w/v) BSA in TBST buffer and then incubated with specific primary antibodies in optimized dilution ratios, followed by incubation with horseradish peroxidase-conjugated secondary antibodies. Immunoreactive bands were visualized by the enhanced chemiluminescence detection system and the intensity of bands was quantified using a model GS-700 Imaging Densitometer (Bio-Rad).

### $Ca^{2+}$ concentration measurement
Hela or HepG2 cells were cultured in black 96-well plates in DMEM culture medium with high glucose overnight (5% $CO_2$ and 37 °C). The culture medium was then removed, and cells were loaded with 5 µM fura 2-AM in Hanks balanced salt solution (HBSS) for 30 min at 37 °C. After removing the loading buffer, cells were washed with HBSS and

excited alternately at 340 and 380 nm using SpectraMax iD3 Multi-Mode Microplate Reader (Molecular Devices). Background fluorescence at 340 and 380 nm (ex) was obtained in wells containing unloaded cells in HBSS buffer and was subtracted from raw fluorescent intensity at the corresponding wavelengths. Background corrected fluorescence at 340 nm (ex) was divided by that at 380 nm (ex) to obtain ratio 340:380. The intracellular $Ca^{2+}$ levels were estimated as the ratio of the signals ($F_{340}/F_{380}$). $CrCl_3$ at different concentrations were acutely added into the well and experiments were performed in triplicate.

## Mitochondrial morphology analysis

HepG2 cells were subjected to high glucose stress to evaluate the effect of Cr(III). Briefly, HepG2 cells with or without pre-treatment of siRNAs were cultured in high level of D-glucose (40 mM). $CrCl_3$ was subsequently added to the culture medium with final concentrations of 10 μM, 100 μM and 1 mM. After incubation for 12 h, culture medium was discarded and the samples were incubated in HBSS preloaded with MitoTracker® Red CMXRos to a final concentration of 50 nM for 30 min, subsequently washed three times by HBSS and subjected to confocal imaging (n = 10). HepG2 cells that are under normal condition (5.6 mM D-glucose) were used as a control. Quantitative analysis of mitochondrial morphology was conducted by computer-assisted analysis[71] using MetaMorph software (version Premier 7.7.0.0) to calculate the values of form factor (FF) and aspect ratio (AR). Acquired images were firstly processed with a top hat morphology filter with square of 7 pixels, then measured with Integrate Morphometry Analysis to calculate the shape factor and elliptical form factor, where FF equals to the reciprocal of shape factor and AR equals to the elliptical form factor. While AR factor indicates how elliptical the mitochondria are, FF value represents the level of mitochondrial branching.

## Mitochondrial membrane potential and ROS measurement

HepG2 cells were cultured on clear bottom and dark sided 96-well microplates in 100 μL DMEM medium ($2 \times 10^5$ cells per mL). After 24 h, the cells were subjected to high glucose and $CrCl_3$ treatment. After incubation for 24 h, assays of JC-1 Mitochondrial Membrane Potential Assay Kit (Thermo Fisher Scientific) and MitoSOX™ Red (Thermo Fisher Scientific) were used to measure mitochondrial membrane potential and ROS respectively according to the manufacture's protocols. In brief, the cells were stained with JC-1 (10 μM) or MitoSOX™ Red (10 μM) at 37 °C for 30 min. After staining, the cells were washed with PBS for three times and measured by SpectraMax iD3 Multi-Mode Microplate Reader (Molecular Devices). Both ROS and membrane potential experimental data were corrected by subtracting the average fluorescence from baseline measurements.

## Glucose consumption

HepG2 cells were cultured in 96-well microplates in 100 μL DMEM medium ($2 \times 10^5$ cells per mL) for 24 h and then exposed to 1 μM insulin for another 24 h. ATP5B in HepG2 cells was knocked down via transfection of ATP5B siRNA using Lipofectamine RNAiMAX reagent, while HepG2 cells transfected with scrambled siRNA were used as a negative control. The IR cells were then subjected to treatment of 100 μM $CrCl_3$ for 8 h. The glucose concentration in the medium was determined by the Glucose Assay Kit (Sigma-Aldrich) based on the glucose oxidase/peroxidase enzymatic reactions. The amount of glucose consumption was calculated by subtraction of glucose concentrations between the cell plated wells and the blank wells. The cells were collected and lysed. And the total protein amount of each well was measured using a BCA protein assay kit. The glucose consumption was normalized to the protein content.

## Animal study

All experiments were performed in accordance with the Animal Use Protocol approved by Committee of Scientific Research in Research Centre for Eco-Environmental Sciences, Chinese Academy of Sciences. C57BL/Ks (BKS) wild-type and type II diabetes model mice db/db were purchased from Cavens Lab Animal, Inc. of age 5 weeks old at the beginning of experiment. The inbred mouse strain BKS carrying a mutation of the leptin receptor *lepr.* (BKS-db) is a classic mouse model of type 2 diabetes. All animals are male mice. All the mice were housed under a SPF condition (12-h light/dark cycle, 40–70% relative humidity, between 22 and 24 °C). Animals were randomly allocated to treatment groups prior to collection of any data. Both diabetic *db/db* and WT mice were feed without or with $Cr^{3+}$ (single oral dose of 10 mg/kg/day) through drinking water systems from the fifth week (n = 6 for each group). The food consumption, body weight, and blood glucose changes were determined regularly with the total time lasted 11 weeks. At the end of the treatment, the mice were sacrificed and tissues from different organs (e.g., liver, kidney, and muscle et al) were collected and processed for the determination of Cr contents, ATP synthase activity, AMP/ATP ratio, and phosphorylated AMPK and ACC. The experimental details of these tests are identical to that of the cell-based studies.

## Statistical analysis

Unless specified, all experiments were subjected to three biological replicates and two technique replicates. A two-tailed t test was used for all comparisons between two groups. Data are presented as mean ± SEM. All comparisons between the control group and the treated group were performed on two-sided student t test. *$p < 0.05$, **$p < 0.01$, ***$p < 0.001$ and ****$p < 0.0001$. NS, not significant ($p > 0.05$). All analyses were carried out using GraphPad Prism software (version 6.01, GraphPad Software, Inc., La Jolla, CA).

## Reporting summary

Further information on research design is available in the Nature Portfolio Reporting Summary linked to this article.

# Data availability

The mass spectrometry proteomics data have been deposited to the ProteomeXchange Consortium via the PRIDE partner repository with the dataset identifier PXD027305. The protein-protein interaction network of Cr(III)-associated proteins was obtained and visualized as confidence view via STRING 9.1 (www.string-db.org). All data supporting the findings of this study are available in a publicly accessible repository or in the Source Data file. The source data are provided with this paper. A reporting summary for this article is available as a Supplementary Information file. Source data are provided with this paper.

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

## Acknowledgements

We acknowledge the assistance of Li Ka Shing Faculty of Medicine Faculty Core Facility (HKU), the Norman & Cecilia Yip Foundation of the University of Hong Kong, and financial supported by the Research Grants Council of Hong Kong (17307017 and 2122-7S04) to H.S., and National Science Foundation of China (22193052) to L.H.

## Author contributions

H.W., L.H. and H.L. contributed equally to this work. H.S. conceived the idea and obtained funding for the projects. H.W., L.H., Y.-T.L., H.L., X.X., G.J., M.-L.H., A.X. and H.S. designed the experiments; H.W., Y.-T.L., L.H., X.X., H.C. and X.W., Z.C., Y.-Y.C. and Q.W. performed the experiments; H.W., Y.-T.L., L.H., H.L., Q.Z. and H.S. analysed the data; H.W., L.H., H.L., Y.-T.L. and H.S. wrote the manuscript with the input from all. All authors discussed the results and commented on the manuscript.

## Competing interests

The authors declare no competing interests.
