## [Peer Review File · Nature Communications]

Mitochondrial ATP synthase as a direct molecular target of chromium(III) to ameliorate hyperglycaemia stressREVIEWER COMMENTS

Reviewer #1 (Remarks to the Author):

I am delighted to see this longstanding scientific issue of the biological role of chromium(III) being addressed by an experienced investigator and his group. It is an immensely important aspect of basic science and its implications for human health, and it is exactly what is needed: the identification of molecular targets, an endeavour that has met with rather limited success in the last 70 years. Chromium supplements are a huge market world-wide (sold for weight loss, muscle building). With controversies about their mode of action it is a rather questionable gambling with our health. Thus, the work addresses a major health issue that needs to be carefully evaluated for either the protection of large populations or the benefit for them concerning antidiabetic actions of chromium(III).

The authors address chromium biology with a new approach of chemical biology and an impressive number of biochemical, molecular biological, and animal experiments. They use a fluorescent probe as a ligand of chromium(III). The probe can be photo-activated for covalent labelling. They identify eight labelled proteins, demonstrate that mitochondria are targeted preferentially, and characterize the beta subunit of mitochondrial ATP synthase (ATP5B) as a major target of chromium with Thr223 and Glu242 as the binding site and a nucleotide analogue as ligand. They come up with a plausible explanation of the antidiabetic action of chromium through a pathway of increased AMP/ATP ratios affecting the key enzyme AMPK based on their experiments with diabetic db/db and WT mice. The mice obtain 10 mg/kg (10 ppm) chromium in their drinking water. It is 100 times higher than the EPA standard of 0.1 mg/L. I am not so much concerned about the higher concentration because one starts frequently with higher concentrations in experimental systems to establish an effect, and the authors are investigating a pharmacological action. They also report another important finding: there is less mitochondrial fragmentation (fission) with chromium, relating chromium to the ageing process.

The use of the chemical probe is a limitation, but I am also not so much concerned about it because it is a new approach and in the right direction, namely clarifying a drug approach that may be important for treating diabetes type 2, for which new medication is direly need in view of the endemic proportions of the disease in many parts of the world. The limitation is that we do not know what the biological transformation (metabolism) of chromium(III) is. A peptide complex or even insulin as a ligand has been suggested. The chemical probe may target chromium specifically to mitochondria. I wonder whether the authors have performed some control experiments to address the translocation of the probe without chromium.

Some additional comments on the chemistry of the probe are necessary. The Figure shows tetrahedral coordination, but the preferred chemistry of chromium(III) is octahedral. Are the remaining coordination sites occupied by water or chloride ions? Is there exchange of these additional ligands such as in cis-platinum compounds? A characteristic feature of this valence state of chromium is a very high kinetic stability. What is the half-life of the complex?

The possible oxidation of chromium(III) is not addressed. Does it remain chromium(III) and is the inhibition of thiol oxidoreductases indeed due to chromium(III) or due to some chromate formed. The

authors may consider mentioning that chromate inhibition of the enzymes has been reported (e.g. C.R.Myers, *Free Radic Biol Med* 52, 2091, 2012). Cr(IV) is clearly rather toxic. A sentence may be needed to educate the reader about the important differences of the major two oxidation states in biology, the redox properties of chromium, or controversies about oxidation of Cr(III) in cells.

In the older literature, an effect of chromium on mitochondria has been reported: Interaction of insulin and chromium(III) on mitochondrial swelling (*Am. J. Physiol.* 204 (1963) 1028).

The reference on transition metal signalling, ref. 51, seems inappropriate. There is no connection of this work to transition metal ion signalling, which requires generating a signal and transmitting it to a target. The exchange-inert nature of chromium(III) makes a role in signalling highly unlikely.

It will be very interesting to see in the future how these pharmacological effects can be exploited and whether they relate to possible nutritional effects.

Reviewer #2 (Remarks to the Author):

This manuscript entitled “Mitochondrial ATP synthase as the first direct molecular target of chromium (III) to ameliorate hyperglycemia stress” by Wang H, et al. investigated that mitochondrial ATP synthase ameliorates hyperglycemia in mice. Authors showed that Cr³⁺ binds to ATP synthase at its subunit beta via the catalytic residues of Thr213/Glu242, and the nucleotide in the active site. Such a binding suppresses ATP synthase activity, leading to activation of AMPK and rescuing mitochondria from hyperglycemia-induced fragmentation. The mode of action of Cr³⁺ in cells also holds true in type II diabetic mice. Although this work is interesting to seek a new function of chromium (III) in the regulation of ATP synthase and its metabolic functions, it is still unclear how chromium (III) regulates glucose metabolism in db/db mice due to the lack of evidence from the tissue-specific knockout mice. There is also the lack of mechanistic studies to show the role of Cr³⁺ in glucose metabolism and insulin resistance in the context of type 2 diabetes in vivo. Some mouse models of obesity-induced insulin resistance and diabetes such as HFD-induced insulin resistant mice are not described in this study. Hence, experimental design should be improved to support the conclusion.

There are major comments:

1. In Fig. 1, authors showed that phosphorylation of AMPK was dose-dependently increased in Cr³⁺ treated HepG2 cells. In Fig. 1, none of the presented experiments provides evidence—whether the effect of Cr³⁺ only occurs in hepatocytes or it also occurs in other metabolic cells such as adipocytes or beta cells. It is unclear the tissue-specific effect of Cr³⁺ in this study.

2. In Fig. 4. Authors showed that Cr³⁺ reduces hyperglycemia in db/db mice, but the effects of Cr³⁺ on hepatic gluconeogenesis are not investigated in db/db mice.

3. In Fig. 4, authors have not described why db/db mice were selected in this study. It is unclear why a HFD mouse model is not used in order to investigate the effect of Cr³⁺ on obesity-induced insulin resistance and hyperglycemia.

4. In Fig. 2, it is unclear how Cr³⁺ increases the activity of AMPK in the liver of db/db mice or in vitro. It is unclear which mechanism (s) such as the upstream kinases of AMPK—LKB1 or CaMKKβ—are involved in the regulation of AMPK by Cr³⁺.

5. Authors should include additional experiments to determine whether altered activity of either ATP synthase or AMPK affects the action of Cr³⁺ on glucose metabolism in insulin resistant or diabetic mice.

Reviewer #3 (Remarks to the Author):

Reviewer's comments on the manuscript by Wang et al. "Mitochondrial ATP synthase as the first direct molecular target of 3 chromium(III) to ameliorate hyperglycaemia stress"

In this work, Wang et al. address the question of whether Cr³⁺-binding proteins exist in mammalian cells. For this they used a fluorophore probe coupled to a chelator that binds Cr³⁺ ions. Cells were incubated with this probe, whereafter proteins were extracted and separated by 1D and 2D gel electrophoresis. Proteins that showed fluorescence were considered to bind the probe. These were identified by MALDI-ToF-MS. By this method the authors identified the two mitochondrial proteins ATP5B and ATP5L, and the proteins Hsp60, TXN, PRDX1, COMT, CLIC1 and H3F3A. The authors performed a variety of studies (ICP-MS, competition assays and cellular thermal-shift assays) to confirm their finding that these proteins indeed bind Cr³⁺. The authors found a possible mechanism of how Cr³⁺ might act in the cell. Finally, they correlate their findings with hyperglycaemia.

I was explicitly asked to judge the MS data, which I will do first in the following comments.

The authors used MALDI-ToF-MS to identify the main proteins within gel spots or gel regions that show fluorescence. Although MALDI-ToF-MS is not sensitive enough to identify all proteins of a gel spot or a gel region when compared with ESI-LC-MS, the data presented here reflect the most abundant protein species found in their analyses. In this respect the data seem to be valid. Additionally, ICP-MS confirmed binding of Cr³⁺ to (or its association with) protein ATP5B expressed in vitro.

Regrettably, I have substantial criticism of the experiments centred around the binding/association of Cr³⁺ with cellular proteins. This makes me very reluctant to support the publication of this work:

1. There seems to be a complete absence of any control, i.e. incubation of cells with the probe without Cr³⁺. Thus, the authors do not appear able to rule out the possibility that the fluorophore interacts with the proteins without Cr³⁺.
2. I see no experiment that demonstrates the specificity of the Cr³⁺ binding/association, i.e. using the same probe chelated e.g. Fe, Mo, Mn, Cu, Ni, etc. at the same concentrations, which are in my opinion far from having any physiological relevance. See also point 1.
3. It is not clear whether the probe with its loaded chelator binds e.g. to phosphate groups or to other residues (chelated Fe and Ni do so).
4. The authors should have investigated whether the chelated Cr³⁺ in the probe can be displaced by any of the above atoms/cations that might be present in a cell.
5. All the experiments with the selected proteins shown an effect of Cr³⁺. It would be more convincing, e.g. in the case of the thermal-shift assay, if proteins had also been selected that do not show such an effect; this would have ruled out the possibilities (i) that this effect was not caused by the probe itself and (ii) that this effect was not simply caused by the incubation of cells with the probe. (A minor point in this respect: the legends for Ctr and Cr³⁺ in the panel showing the effect on Hsp60 seem to have been swapped.)

All in all, on the basis of the points listed above, I cannot recommend that this paper be published in Nature Communications.

Reviewer #4 (Remarks to the Author):

This manuscript by Wang and colleagues describes the mechanism by which Cr³⁺ suppresses the effects of hyperglycemia stress by inhibiting the mitochondrial ATP synthase. The authors used a clever fluorescence-based approach coupled with MS to identify Cr³⁺-binding proteins. The top candidates included two subunits of the mitochondrial ATPase and the mitochondrial chaperone HSP60. Fluorescence microscopy showed that most Cr³⁺ signal localizes to mitochondria, and AMPK activity measurement indicated induced activation due to a defective mitochondrial ATP synthase. The specific effects under hyperglycemia conditions may be explained by the fact that more Cr³⁺ is uptake by the cells and mitochondria in these conditions. Cr³⁺ is shown to bind specifically to ATPase subunit ATP5B, and the silencing of this subunit induces similar effects on AMPK activation, AMP/ATP ratio, and restoration of hyperglycemia-induced mitochondrial fragmentation. Finally, the authors show that Cr³⁺ feeding is able to protect against hyperglycemia stress in diabetic mice.

The manuscript is technically and conceptually sound. However, a few questions remain to better understand the mechanism proposed.

1- Cr³⁺ dependent inhibition of the mitochondrial ATPase is expected to alter two key parameters such as the mitochondrial membrane potential and the generation of superoxide. The contribution of these parameters needs to be measured and discussed.

2- It is intriguing both that hyperglycemia induces mitochondrial fragmentation and that Cr³⁺ restores it. The mechanism by which Cr³⁺-binding to ATP5B impacts mitochondrial dynamics needs to be further discussed.

3- Since the mitochondrial ATPase plays an important role in cristae formation/stabilization, it would be important to test whether Cr³⁺-binding to ATP5B prevents the required ATPase oligomerization involved in the cristae formation process.

Re: “Mitochondrial ATP synthase as the first direct molecular target of chromium(III) to ameliorate hyperglycaemia stress” (NCOMMS-21-25340)

Reviewer #1 (Remarks to the Author):

I am delighted to see this longstanding scientific issue of the biological role of chromium(III) being addressed by an experienced investigator and his group. It is an immensely important aspect of basic science and its implications for human health, and it is exactly what is needed: **the identification of molecular targets, an endeavour that has met with rather limited success in the last 70 years.** Chromium supplements are a huge market world-wide (sold for weight loss, muscle building). With controversies about their mode of action it is a rather questionable gambling with our health. Thus, the work addresses a major health issue that needs to be carefully evaluated for either the protection of large populations or the benefit for them concerning antidiabetic actions of chromium(III).

The authors address chromium biology with a new approach of chemical biology and an impressive number of biochemical, molecular biological, and animal experiments. They use a fluorescent probe as a ligand of chromium(III). The probe can be photo-activated for covalent labelling. They identify eight labelled proteins, demonstrate that mitochondria are targeted preferentially, and characterize the beta subunit of mitochondrial ATP synthase (ATP5B) as a major target of chromium with Thr223 and Glu242 as the binding site and a nucleotide analogue as ligand. They come up with a plausible explanation of the antidiabetic action of chromium through a pathway of increased AMP/ATP ratios affecting the key enzyme AMPK based on their experiments with diabetic db/db and WT mice. The mice obtain 10 mg/kg (10 ppm) chromium in their drinking water. It is 100 times higher than the EPA standard of 0.1 mg/L. I am not so much concerned about the higher concentration because one starts frequently with higher concentrations in experimental systems to establish an effect, and the authors are investigating a pharmacological action. They also report another important finding: there is less mitochondrial fragmentation (fission) with chromium, relating chromium to the ageing process.

The use of the chemical probe is a limitation, but I am also not so much concerned about it because it is a new approach and in the right direction, namely clarifying a drug approach that may be important for treating diabetes type 2, for which new medication is direly need in view of the endemic proportions of the disease in many parts of the world. The limitation is that we do not know what the biological transformation (metabolism) of chromium(III) is. A peptide complex or even insulin as a ligand has been suggested. The chemical probe may target chromium specifically to mitochondria. I wonder whether the authors have performed some control experiments to address the translocation of the probe without chromium.

[Response to Reviewer #1]

We truly appreciate the reviewer’s favorable comments and helpful suggestions on the present study as well as recognition on the major merits of our work. These suggestions are immensely useful for us to improve the manuscript.

The probe used in this study is a new addition of our previously reported metal-tunable probes (M^{n+} -NTA-AC or renamed as M^{n+} -TRACER) (*Acc Chem Res*, 2019, 52, 216; *Metallomics*, 2018, 10, 77; *Chem Sci*, 2019, 10, 6099; *Chem Sci*, 2017, 8, 4626), which have been extensively utilized to track metal-binding proteins in cells. To avoid misunderstanding and be consistent with our previous publications, we changed Cr^{3+} -FL into Cr^{3+} -NTA-AC in the revised version.

In our previous studies, we showed that the probe without chromium(III) (or other metals) cannot enter the cells, thus the concern about the translocation of the probe could be alleviated. Moreover, we showed that different set of proteins were identified by different M^{n+} -NTA-AC probes, suggesting the metal instead of ligand-oriented binding to the target proteins.

Some additional comments on the chemistry of the probe are necessary. The Figure shows tetrahedral coordination, but the preferred chemistry of chromium(III) is octahedral. Are the remaining coordination sites occupied by water or chloride ions? Is there exchange of these additional ligands such as in cis-platinum compounds? A characteristic feature of this valence state of chromium is a very high kinetic stability. What is the half-life of the complex?

[Response 1] We truly appreciate the reviewer's insightful comments. The remaining ligands are likely chloride ions at the very beginning since we used CrCl_3 to synthesize the probe, however, the chloride ions could be replaced by H_2O eventually as found for $\text{CrCl}_3 \cdot 6\text{H}_2\text{O}$ (*Physiol Rev*, 1969, 49, 163). We agree with the reviewer that the Cr^{3+} is highly kinetic stable *in vitro*. However, we found inside cells, the probe Cr^{3+} -NTA-AC could rapidly label proteins (within 30 min), and the reactivity of the complex might be very different *in vitro* and inside cells. The probe is highly stable and will bind to the proteins in cells without dissociation of Cr(III) from the probe as we revealed from a ternary complex of protein-metal(Fe^{3+})-NTA-AC. Thus, the half-life of the probe won't affect the labelling.

The possible oxidation of chromium(III) is not addressed. Does it remain chromium(III) and is the inhibition of thiol oxidoreductases indeed due to chromium(III) or due to some chromate formed. The authors may consider mentioning that chromate inhibition of the enzymes has been reported (e.g. C.R.Myers, *Free Radic Biol Med* 52, 2091, 2012). Cr(IV) is clearly rather toxic. A sentence may be needed to educate the reader about the important differences of the major two oxidation states in biology, the redox properties of chromium, or controversies about oxidation of Cr(III) in cells.

[Response 2] We truly appreciate the reviewer's very helpful suggestion. Under the reducing intracellular environment, Cr(III) is the most likely species and remained quite stable in biological systems (*The nutritional biochemistry of Cr(III)*, 2019, p341). The Cr(III)/Cr(II) redox potential is around -0.42 V at 25 °C (*Standard Potentials in Aqueous Solution*, 1985, p19). Under certain circumstances, Cr(III) might be reduced to Cr(II), which is very unstable in biological systems. However, this reaction is speculative and has been proposed to occur with Cr(III) complexes containing aromatic ligands only (e.g., bipyridyl and phenanthroline) and in the presence of strong biological reductants, such as ascorbate (*The nutritional biochemistry of Cr(III)*, 2019, p342).

We believe that probe Cr^{3+} -NTA-AC should be very stable and is unlikely to be oxidized given the tight binding of Cr^{3+} to NTA. Moreover, pretreatment of CrCl_3 to the identified proteins (ATP5B, ATP5L and Hsp60) resulted in reduced fluorescence intensity, indicative of competitive binding of CrCl_3 with Cr^{3+} -NTA-AC, which further validates the proteins identified by the probe is Cr(III)-binding proteins.

Nevertheless, as suggested by this reviewer, we have added a sentence to mention that Cr(IV) inhibition of the enzymes has been reported (page 6, line 2-3). We also added a sentence to discuss the differences of Cr(III) and Cr(IV) in biology and cited the reference as suggested in the discussion part (page 6, line 1-2).

In the older literature, an effect of chromium on mitochondria has been reported: Interaction of insulin and chromium(III) on mitochondrial swelling (*Am. J. Physiol.* 204 (1963) 1028).

[Response 3] This reference has been cited.

The reference on transition metal signalling, ref. 51, seems inappropriate (remove this). There is no connection of this work to transition metal ion signalling, which requires generating a signal and transmitting it to a target. The exchange-inert nature of chromium(III) makes a role in signalling highly unlikely.

[Response 4] This reference has been deleted in the revised version as suggested.

It will be very interesting to see in the future how these pharmacological effects can be exploited and whether they relate to possible nutritional effects.

[Response 5] We agree with the reviewer and hope our study could lay a foundation to extend the nutritional application of Cr(III).

Reviewer #2 (Remarks to the Author):

This manuscript entitled “Mitochondrial ATP synthase as the first direct molecular target of chromium (III) to ameliorate hyperglycemia stress” by Wang H, et al. investigated that mitochondrial ATP synthase ameliorates hyperglycemia in mice. Authors showed that Cr³⁺ binds to ATP synthase at its subunit beta via the catalytic residues of Thr213/Glu242, and the nucleotide in the active site. Such a binding suppresses ATP synthase activity, leading to activation of AMPK and rescuing mitochondria from hyperglycemia-induced fragmentation. The mode of action of Cr³⁺ in cells also holds true in type II diabetic mice. Although this work is interesting to seek a new function of chromium (III) in the regulation of ATP synthase and its metabolic functions, it is still unclear how chromium (III) regulates glucose metabolism in db/db mice due to the lack of evidence from the tissue-specific knockout mice. There is also the lack of mechanistic studies to show the role of Cr³⁺ in glucose metabolism and insulin resistance in the context of type 2 diabetes in vivo. Some mouse models of obesity-induced insulin resistance and diabetes such as HFD-induced insulin resistant mice are not described in this study. Hence, experimental design should be improved to support the conclusion.

[General Response to Reviewer #2]

We truly thank this reviewer’s helpful suggestions to improve the manuscript. We have performed more experiments as suggested and added more discussion (page 6-7) in the revised manuscript, in which we clearly elaborated the major merits of our study which was highly recognized by other reviewers (e.g. **Rev 1#**: “I am delighted to see this longstanding scientific issue of the biological role of chromium(III) being addressed by an experienced investigator and his group. It is an immensely important aspect of basic science and its implications for human health, and it is exactly what is needed.....the identification of molecular targets, an endeavor that has met with rather limited success in the last 70 years.”)

We apologize for not providing enough background information regarding the effects of chromium(III) on glucose metabolism and have added relevant information and references in the revised version in the introduction part. Considering extensive studies on the antidiabetic effects of Cr(III) in mouse models, the major merit of this work lies in the molecular mechanism of action of Cr(III), which has been concealed for over fifty years. In this work, we used chemical biology approach to first identify molecular targets of Cr(III) in liver cells and to further validate these targets, which provides fundamental basis for subsequent study of the role of Cr(III). As the identified protein targets are mainly enriched in

ATP synthesis, we further investigated how Cr(III) inhibits ATP synthase and found that Cr(III) binds to ATP synthase at beta subunit, leading to elevated AMP/ATP ratio, consequently resulting in activation of AMPK. In this sense, the antidiabetic activity is similar to metformin, the first line-antidiabetic drug. It is well known that activation of AMPK stimulates glucose uptake and fatty acid oxidation in skeletal muscle and heart, but inhibits fatty acid synthesis in liver and lipogenesis in adipose tissue (*Diabetes Metab Syndr Obes* 2014, 7, 241; *Am J Physiol* 1999, 277, E1) and silencing AMPK abolished the effect of Cr(III) (*J Nutr Biochem* 2014, 25, 565). It is highly possible that Cr(III) does not directly regulate glucose metabolism or insulin activity, instead indirectly activating AMPK through inhibition of ATP synthase, consequently regulating glucose metabolism. We have performed additional experiments to confirm this (**Fig. 4a-c and Supplementary Fig. 8**).

Since the first report that addition of Cr(III) salts to a rat diet increased blood glucose removal, there have been extensive studies on how Cr(III) regulates glucose metabolism and enhances insulin signaling. Till now, the molecular mechanism remains unknown in spite of enormous efforts. Based on our study, such an effect is indirect consequence of Cr(III) inhibition of ATP synthase. Given that multiple protein targets of Cr(III) is identified, it is likely that the role of Cr(III) is far more from just anti-diabetic, other pharmacological application is highly possible such as anti-neurodegeneration, anti-aging. Thus, this study opens a new horizon for further exploration of pharmacologic and medicinal applications of Cr(III).

Given that we addressed the longstanding issue on the molecular mechanism of action of Cr(III), we believe that this study deserves to be published in *Nature Communications*.

There are major comments:

1. In Fig. 1, authors showed that phosphorylation of AMPK was dose-dependently increased in Cr³⁺ treated HepG2 cells. In Fig. 1, none of the presented experiments provides evidence—whether the effect of Cr³⁺ only occurs in hepatocytes or it also occurs in other metabolic cells such as adipocytes or beta cells. It is unclear the tissue-specific effect of Cr³⁺ in this study.

[Response 1] We truly appreciate the reviewer's helpful suggestion. Indeed, apart from HepG2 cells, we also selected C2C12 muscle cells for validation and observed a similar Cr³⁺-mediated inhibition of ATP synthase and activation of AMPK and ACC in C2C12 cells (**Supplementary Fig. 6, 7**), which is consistent with previous report on Cr³⁺-mediated activation of AMPK and ACC in rat L6 muscle cells (*J Nutr Biochem*, 2014, 25, 565). In addition, there are also reports on Cr³⁺-mediated promotion of mitochondrial biogenesis and lipid metabolism in 3T3-L1 adipocytes via activation of AMPK and ACC (*Clin Exp Pharmacol Physiol*, 2009, 36, 843; *PLoS One*, 2015, 10, e0131930; *Arterioscler Thromb Vasc Biol*, 2011, 31, 1139). Thus, the effects of Cr³⁺ are not cell or tissue specific, though the level of those effects vary among different cells or tissues.

2. In Fig. 4. Authors showed that Cr³⁺ reduces hyperglycemia in db/db mice, but the effects of Cr³⁺ on hepatic gluconeogenesis are not investigated in db/db mice.

[Response 2] We truly thank the reviewer's helpful suggestion. As suggested, we further measured the effects of Cr³⁺ on hepatic gluconeogenesis in both WT and *db/db* mice. As shown below (**Supplementary Fig. 15** in the revised manuscript), the treatment of Cr³⁺ led to decrease of the abundance of glucose-6-phosphatase (G6Pase) and phosphoenolpyruvate carboxykinase (PEPCK) to

0.72 ± 0.11 and 0.70 ± 0.08 (the control group was set as 1) in the liver of *db/db* mice, whereas no significant suppression of G6Pase and PEPCK was observed in Cr³⁺-treated WT mice, which is consistent with previous report (*J Ocean Univ China*, 2011, 10, 336).

We have included the new data and related references in the revised manuscript (page 5, line 55-58).

Supplementary Fig. 15 CrCl₃ suppresses hepatic gluconeogenesis in *db/db* mice but not normal mice (WT). The effect of Cr³⁺ treatment on the expression level of hepatic gluconeogenesis enzymes in *db/db* (a) and (b) WT mice (n = 6). The results are shown as mean ± SEM. The asterisks denote significant difference of different groups (**P* < 0.05, ***P* < 0.01, ****P* < 0.001 and *****P* < 0.0001). NS, not significant (*P* > 0.05).

3. In Fig. 4, authors have not described why *db/db* mice were selected in this study. It is unclear why a HFD mouse model is not used in order investigate the effect of Cr³⁺ obesity-induced insulin resistance and hyperglycemia.

[Response 3] We thank the reviewer's insightful comment. We agree with the reviewer that a rational of selection of *db/db* mice in this study is necessary and appropriate models are essential to evaluate various compounds that have therapeutic effects on type 2 diabetes.

The *db/db* mouse model of leptin deficiency is currently the most widely used mouse model of type 2 diabetes mellitus (T2DM). This mouse has a mutation in the gene encoding the leptin receptor, and leptin deficiency confers susceptibility to obesity, insulin resistance, and T2DM (*Curr Opin Nephrol Hypertens*, 2011, 20, 278). The *db/db* mice with the C57BLKS/J strain are widely used, and therefore we used in the current study (*J Diabetes Res*, 2013, 2013, 797548). The C57BLKS/J-*db/db* diabetic mouse, similar to patients with type 2 diabetes mellitus and characterized by obesity, hyperglycemia, and hyperinsulinemia, is a well-studied murine model to examine the pathogenesis and treatment of diabetes (*Diabetologia*, 1975, 11, 431; *Metab Clin Exp*, 2000, 49, 22).

Indeed, the antidiabetic effects of Cr(III) has been investigated in *db/db* mice (*J Trace Elem Med Biol*, 2020, 62, 126606; *Bioorg Chem*, 2019, 88, 102942). These studies showed that Cr(III) can effectively decrease the blood glucose level, improve blood lipid metabolism, and increase insulin sensitivity without acute hypoglycemic side effect in *db/db* mice. Besides, Cr(III) also activates AMPK signaling

pathway to regulate glycogen synthesis, gluconeogenesis and lipid metabolism in the liver, and attenuate the hyperglycemic symptom in *db/db* mice (*J Ocean Univ China*, 2011, 10, 336).

In this study, we aim to address the molecular mechanism underlying the antidiabetic effect (*e.g.*, ameliorate hyperglycaemia and activation of AMPK) of Cr(III). With the *db/db* mouse model that has been demonstrated with the same pharmacological effects of Cr(III), we could readily compare our results with the precious findings, we therefore choose the *db/db* mice model. We have added a description to elucidate why *db/db* mice were selected as suggested (page 6, line 26-37).

4. In Fig. 2, it is unclear how Cr^{3+} increases the activity of AMPK in the liver of *db/db* mice or in vitro. It is unclear which mechanism(s) such as the upstream kinases of AMPK—LKB1 or CaMKK β —are involved in the regulation of AMPK by Cr^{3+} .

[Response 4] We truly appreciate the reviewer's very helpful suggestion, and further examined the mechanism of Cr(III)-mediated activation of AMPK.

According to a previous report (*Int J Obes*, 2008, 32, S55), the key step in the activation of AMPK is its phosphorylation on threonine 172 (Thr172) within its catalytic subunit. Liver kinase B1 (LKB1) and Ca^{2+} /calmodulin-dependent protein kinase kinase- β (CaMKK β) have been identified as two kinases that phosphorylate Thr172. In response to energy stress, resulting increase in AMP:ATP ratio, LKB1 phosphorylates AMPK, whereas the activation of AMPK by CaMKK β is initiated by an increase in intracellular Ca^{2+} and does not respond to change in the ratio of AMP/ATP (*Cell Metab*, 2005, 2, 9; *Cell Metab*, 2005, 2, 21; *J Biol Chem*, 2005, 280, 29060).

Based on our results, the AMP/ATP ratio was significantly increased after HepG2 cells were supplemented with Cr^{3+} under hyperglycaemia stress, so we first examined whether LKB1 was involved in the regulation of AMPK by Cr^{3+} . Hela cells, a LKB1 deficient cell line, was employed to study Cr^{3+} -induced AMPK activation. AICAR, an AMP analogue activates AMPK through AMP- and LKB1-dependent mechanism and has no effect on AMPK α phosphorylation in Hela cells (*J Biol Chem*, 2003, 2, 28; *PLoS One*, 2012, 7, e47900), was used as a negative control.

As shown below **Supplementary Fig. 8a** in the revised version (*below*), we investigated whether Cr^{3+} could activate AMPK and its key target ACC in Hela cells under hyperglycaemia condition by measuring AMPK Thr172 phosphorylation and ACC Ser-79 phosphorylation through western blotting. Compared with the control group, no obvious increase in AMPK and ACC phosphorylation was observed in Hela cells treated with increasing concentrations of CrCl_3 , which is different from the observation in HepG2 cells, indicating that LKB1 is involved in the activation of AMPK and ACC upon Cr^{3+} treatment in HepG2 cells.

Since AMPK can be activated by two distinct mechanisms, *i.e.*, AMP-dependent pathway and Ca^{2+} -dependent pathway (*Int J Obes*, 2008, 32, S55), we further examined whether CaMKK β is involved in the activation of AMPK by Cr^{3+} . As shown in **Supplementary Fig. 8b-8c** (in the revised version), compared to the control group, treatment of increasing concentrations of CrCl_3 did not induce a significant increase of intracellular Ca^{2+} levels in both Hela and HepG2 cells, indicating that CaMKK β is not involved in the activation of AMPK and ACC under Cr^{3+} treatment.

Taken together, LKB1 but not CaMKK β is involved in Cr³⁺-mediated activation of AMPK in HepG2 cells. These results and discussions have been incorporated into the revised manuscript (page 3, line 58-62 and page 4, line 1-10).

Supplementary Fig. 8a Effect of Cr³⁺ treatment on AMPK (Thr172) and ACC activation in HeLa cells under hyperglycaemia condition as measured by western blotting (n = 3). The results are shown as mean \pm SEM. NS, not significant ($P > 0.05$).

Supplementary Fig. 8 (b-c) Intracellular Ca²⁺ levels in HeLa (b) and HepG2 (c) cells after treatment of different concentrations of CrCl₃ (n = 3). HeLa or HepG2 cells were treated with 1 μM Ca²⁺ indicator fura 2-AM. Intracellular Ca²⁺ levels were estimated as the ratio of the signals (F₃₄₀/F₃₈₀). One representative of three replicates is presented.

5. Authors should include additional experiments to determine whether altered activity of either ATP synthase or AMPK affects the action of Cr^{3+} on glucose metabolism in insulin resistant or diabetic mice.

[Response 5] We truly thank the reviewer's kind suggestion and apologize that we did not provide enough background information on previous studies. The antidiabetic action and improvement of glucose metabolism of Cr(III) has been well established in animal models (both *db/db* insulin resistance mice and high-fat-induced diabetic mice) (*J Nutr*, 2008, 138, 1846; *BioFactor*, 2012, 38, 59; *PLoS One*, 2015, 10, e0125952). In particular, it has also been reported that Cr(III) activates AMPK signaling pathway to improve glucose metabolism and suppress gluconeogenesis enzymes to inhibit hepatic glucose production, which consequently ameliorate insulin resistance and attenuate the hyperglycemic symptom in *db/db* mice (*J Ocean Univ China*, 2011, 10, 336).

In this study, animal studies were designed to validate our results in cells, *i.e.*, Cr(III) inhibits ATP synthase, leading to elevated AMP/ATP ratio, and activation of AMPK. Indeed, the results from our animal study on *db/db* mice are consistent with previous findings in terms of Cr(III)-mediated activation of AMPK and consequent decrease of blood glucose. Furthermore, we also provided additional evidence on Cr(III) inhibition on ATP synthase activity, which subsequently increase AMP/ATP ratio and activate AMPK.

As silencing essential genes such as ATP synthase (ATP5B) in animal is very much time consuming and challenging as well, we performed additional experiments in insulin resistant HepG2 cells under hyperglycemia stress to determine whether altered activity of ATP synthase affects the action of Cr^{3+} on glucose metabolism to address the concerns from the reviewer. ATP5B in HepG2 cells was knocked down *via* transfection of ATP5B siRNA using Lipofectamine RNAiMAX reagent, while HepG2 cells transfected with scrambled siRNA were used as a negative control. As shown in **Fig. 4a**, CrCl_3 promoted glucose consumption of HepG2 cells by 19.7% (from 23.4 to 28.1 mM/g protein). The knock-down of ATP5B also increased the glucose uptake by 23.8% (from 23.4 to 29.0 mM/g protein), while no significant difference of glucose uptake was observed in CrCl_3 treated and non-treated ATP5B null cells, indicating that ATP5B is critical for Cr^{3+} -mediated promotion of glucose uptake. In addition, treatment of CrCl_3 resulted in significant upregulation of glycolysis enzymes glucokinase (GCK) (1.54 ± 0.06) and phosphofructokinase (PFKL) (1.29 ± 0.03) but suppression of gluconeogenesis enzymes G6Pase (0.86 ± 0.01) and PEPCK (0.74 ± 0.02) (**Fig. 4b, c**). Similar effects, *i.e.*, upregulation of GCK (1.55 ± 0.03) and PFKL (1.32 ± 0.04) but suppression of G6Pase (0.77 ± 0.06) and PEPCK (0.71 ± 0.04) were observed in ATP5B null cells compared to negative control cells in the absence of CrCl_3 . However, incubation of the cells with 100 μM Cr^{3+} resulted in no significant increase in the levels of GCK and PFKL and decrease of PEPCK and G6Pase in ATP5B null cells. These data collectively demonstrate that the promotion of glucose metabolism by Cr^{3+} is attributable to Cr^{3+} -induced suppression of ATP synthase activity through targeting ATP5B.

We have incorporated the data and discussion in the revised manuscript (page 5, line 19-31).

Fig. 4 CrCl₃ promotes glycolysis and suppresses gluconeogenesis through targeting ATP5B. (a). Comparison of glucose consumption in HepG2 cells. (b,c) The effect of ATP5B gene silencing on Cr³⁺-mediated regulation of glycolysis and gluconeogenesis enzymes in HepG2 cells (n = 3). The results are shown as mean \pm SEM. The asterisks denote significant difference of different groups (**P* < 0.05, ***P* < 0.01, ****P* < 0.001 and *****P* < 0.0001). NS, not significant (*P* > 0.05).

Reviewer #3 (Remarks to the Author):

Reviewer's comments on the manuscript by Wang et al. "Mitochondrial ATP synthase as the first direct molecular target of chromium(III) to ameliorate hyperglycaemia stress"

In this work, Wang et al. address the question of whether Cr³⁺-binding proteins exist in mammalian cells. For this they used a fluorophore probe coupled to a chelator that binds Cr³⁺ ions. Cells were incubated with this probe, whereafter proteins were extracted and separated by 1D and 2D gel electrophoresis. Proteins that showed fluorescence were considered to bind the probe. These were identified by MALDI-ToF-MS. By this method the authors identified the two mitochondrial proteins ATP5B and ATP5L, and the proteins Hsp60, TXN, PRDX1, COMT, CLIC1 and H3F3A. The authors performed a variety of studies (ICP-MS, competition assays and cellular thermal-shift assays) to confirm their finding that these proteins indeed bind Cr³⁺. The authors found a possible mechanism of how Cr³⁺ might act in the cell. Finally, they correlate their findings with hyperglycaemia.

I was explicitly asked to judge the MS data, which I will do first in the following comments.

The authors used MALDI-ToF-MS to identify the main proteins within gel spots or gel regions that show fluorescence. Although MALDI-ToF-MS is not sensitive enough to identify all proteins of a gel spot or a gel region when compared with ESI-LC-MS, the data presented here reflect the most abundant protein species found in their analyses. In this respect the data seem to be valid. Additionally, IPC-MS confirmed binding of Cr³⁺ to (or its association with) protein ATP5B expressed in vitro.

Regrettably, I have substantial criticism of the experiments centred around the binding/association of Cr³⁺ with cellular proteins. This makes me very reluctant to support the publication of this work:

[General Response to Reviewer #3]

We truly appreciate the reviewer's kind comments and helpful suggestions on the present study.

In fact, the Cr³⁺ probe is a new addition of Mⁿ⁺-NTA-AC (or renamed as Mⁿ⁺-TRACER) probes developed by our group, which have been demonstrated clearly to label His-tagged proteins (*Proc Natl Acad Sci USA*, 2015, 112, 2948) or Fe³⁺-binding proteins (*Metallomics*, 2018, 10, 77), Ga³⁺-binding proteins (*Chem Sci*, 2019,10, 6099) and Bi³⁺-binding proteins (*Chem Sci*, 2017, 8, 4626; *PLoS Biol*, 2018, 16, e2003887) in live cells and these have also been summarized in our recent review (*Ann Rev Biochem*, 2022, 91, 1; *Acc Chem Res*, 2019, 52, 216). These studies have well demonstrated that the Mⁿ⁺-NTA-AC is a mature method for intracellular imaging and identification of metal-binding proteins. We agree with the reviewer that this method is not without limitation and it cannot identify all proteins bind to specific metal, whereas it is one of the best approaches to systematically identify metal-associated proteins so far. Thus, the concern for the methodology could be alleviated. To be consistent with our previous study and avoid confusion, we changed Cr³⁺-FL to Cr³⁺-NTA-AC or Cr³⁺-TRACER in the revised manuscript.

Moreover, the major merit of this work lies in the molecular mechanism of action of Cr(III), which has been concealed for over fifty years. In this work, we used chemical biology approach to first identify molecular targets of Cr(III) in liver cells and to further validate these targets, which provides fundamental basis for subsequent study of the role of Cr(III). As the identified protein targets are mainly enriched in ATP synthesis, we further investigated how Cr(III) inhibits ATP synthase and found that Cr(III) binds to ATP synthase at beta subunit, leading to elevated AMP/ATP ratio, consequently resulting in activation of AMPK and amelioration of hyperglycaemia stress. The major merit is well recognized by other reviewers (“*this longstanding scientific issue of the biological role of chromium(III) being addressed by an experienced investigator and his group. It is an immensely important aspect of basic science and its implications for human health, and it is exactly what is needed: the identification of molecular targets, an endeavour that has met with rather limited success in the last 70 years*”).

1. There seems to be a complete absence of any control, i.e. incubation of cells with the probe without Cr³⁺. Thus, the authors do not appear able to rule out the possibility that the fluorophore interacts with the proteins without Cr³⁺.

[Response 1] We truly appreciate the referee’s kind suggestion. In fact, the Cr³⁺ probe is an excellent addition of Mⁿ⁺-NTA-AC (or renamed as Mⁿ⁺-TRACER) probes developed by this group, which have been demonstrated to label His-tagged proteins (*Proc Natl Acad Sci USA*, 2015, 112, 2948) or Fe³⁺-binding proteins (*Metallomics*, 2018, 10, 77), Ga³⁺-binding proteins (*Chem Sci*, 2019,10, 6099) and Bi³⁺-binding proteins (*Chem Sci*, 2017, 8, 4626; *PLoS Biol*, 2018, 16, e2003887) in live cells and these have also been summarized in our recent reviews (*Ann Rev Biochem*, 2022, 91, 1; *Acc Chem Res*, 2019, 52, 216).

We previously demonstrated that the fluorescence ligand without metal ions cannot enter cells to label proteins. However, upon the fluorescence ligand conjugated with different metal ions, different metal-binding proteins can be tracked, which were further verified in our previous studies. Thus, the proteins identified in this study are orientated by Cr(III) rather than the fluorophore itself. The identified Cr(III) binding proteins are subsequently validated either by *in vitro* or in cellular experiments. Thus, the concern of interaction of fluorophore with the proteins could be alleviated.

In the revised manuscript, we have renamed Cr³⁺-FL to Cr³⁺-NTA-AC to make it consistent with our previous study as suggested.

2. I see no experiment that demonstrates the specificity of the Cr³⁺ binding/association, i.e. using the same probe chelated e.g. Fe, Mo, Mn, Cu, Ni, etc. at the same concentrations, which are in my opinion far from having any physiological relevance. See also point 1.

[Response 2] As indicated above, the Fe, Cu, Ni and Bi binding proteins have been previously identified using the probe chelated with different metal ions, including Ni²⁺-binding proteins (*PNAS* 2015, 112, 2948), Fe³⁺-binding proteins (*Metalomics* 2018, 10, 77), Ga³⁺-binding proteins (*Chem Sci* 2019,10, 6099) and Bi³⁺-binding proteins (*Chem Sci* 2017, 8, 4626; *PNAS* 2015, 112, 3211) in live cells, which have been summarized in our recent review (*Ann Rev Biochem* 2022, 91, 20.1). These studies showed that the metal-binding proteins are specific for metal ions, confirming the metal-orientated binding of the probe to proteins.

3. It is not clear whether the probe with its loaded chelator binds e.g. to phosphate groups or to other residues (chelated Fe and Ni do so).

[Response 3] We truly appreciate the reviewer's constructive comment. We have demonstrated previously that the fluorescence can only be enhanced upon binding of the probe to a target protein (*PNAS* 2015, 112, 2948). The fluorescence is negligible once 'off-target', i.e., when the probe binds low molecular mass ligands (*which is one of the advantages of the probe*).

4. The authors should have investigated whether the chelated Cr³⁺ in the probe can be displaced by any of the above atoms/cations that might be present in a cell.

[Response 4] The concentration of free transition metal in cells is tightly regulated and generally less than μM level (*Curr Opin Chem Biol*, 2008, 12, 222), except for Na⁺, K⁺, and Mg²⁺, yet the binding of these three metals to NTA have been found to be quite weak (logK of K⁺/NTA, Na⁺/NTA and Mg²⁺/NTA are 0.6, 2.15, and 5.46 respectively). Given a concentration of Cr³⁺-NTA-AC (50 μM) was used, these metals could not compete with the Cr³⁺ in Cr³⁺-TRACER even at mM concentrations. Therefore, it is unlikely that Cr³⁺ in the probe could be replaced by other metal ions (*Cr³⁺ is also known to be kinetically inert*) and we did not find interference to Cr(III)-TRACER from the other essential metals.

5. All the experiments with the selected proteins shown an effect of Cr³⁺. It would be more convincing, e.g. in the case of the thermal-shift assay, if proteins had also been selected that do not show such an effect; this would have ruled out the possibilities (i) that this effect was not caused by the probe itself and (ii) that this effect was not simply caused by the incubation of cells with the probe. (A minor point in this respect: the legends for Ctr and Cr³⁺ in the panel showing the effect on Hsp60 seem to have been swapped.)

[Response 5] We truly thank the reviewer's helpful comments. As suggested, we performed the thermal shift assay (CETSA) on GAPDH protein, which is an abundant protein but not labeled by the probe under identical conditions, as a negative control to further validate that the Cr(III)-TRACER could precisely track Cr(III)-binding proteins. Treatment of CrCl₃ resulted in no apparent change of aggregation temperatures (*T_{agg}*, °C), confirming that CrCl₃ indeed does not bind to GAPDH. The result has been incorporated into revised manuscript (**Supplementary Fig. 4**).

Besides, we pre-saturated the proteins with CrCl₃ prior to supplementation of the probe and observed over 10-fold higher fluorescent signals for apo-ATP5L, apo-ATP5B or apo-Hsp60 than those for Cr³⁺

pre-saturated on fluorescence images (Fig. 1e, 1f and supplementary Fig. 4a), indicative of direct competition of the probe and CrCl₃ for binding to the proteins. While in the case of the thermal-shift assay, similar thermal destabilization of these proteins by both CrCl₃ and the probe Cr³⁺-NTA-AC was observed, demonstrating the protein identified by the probe are indeed Cr³⁺ binding proteins.

The legends for Ctrl and Cr³⁺ in the panel showing the effect on Hsp60 have been corrected as suggested.

Supplementary Fig. 4 Cellular thermal-shift assays (CETSA) demonstrating no binding of Cr³⁺ to GAPDH *in cellulo*. All experiments were performed in biological triplicates. The results are shown as mean ± SEM. One representative result of three independent experiments is shown.

All in all, on the basis of the points listed above, I cannot recommend that this paper be published in *Nature Communications*.

[Response 6] With additional experiments and significant revision, we hope we have fully addressed the reviewer’s concerns. Given that this is the first attempt to track Cr(III)-proteome in cells, we believe that we have provided fundamental basis for further investigation of the molecular mechanism of action of Cr(III), which has been concealed for over 70 years. As pointed by the other reviewers, “It is an immensely important aspect of basic science and its implications for human health, and it is exactly what is needed: the identification of molecular targets, an endeavor that has met with rather limited success in the last 70 years (Rev 1#)”.

Given the novel discovery as well as the solid evidence to support the conclusions, we believe that this work deserves to be published in *Nature Communications*.

Reviewer #4 (Remarks to the Author):

This manuscript by Wang and colleagues describes the mechanism by which Cr³⁺ suppresses the effects of hyperglycemia stress by inhibiting the mitochondrial ATP synthase. The authors used a clever fluorescence-based approach coupled with MS to identify Cr³⁺-binding proteins. The top candidates included two subunits of the mitochondrial ATPase and the mitochondrial chaperone HSP60. Fluorescence microscopy showed that most Cr³⁺ signal localizes to mitochondria, and AMPK activity measurement

indicated induced activation due to a defective mitochondrial ATP synthase. The specific effects under hyperglycemia conditions may be explained by the fact that more Cr^{3+} is uptake by the cells and mitochondria in these conditions. Cr^{3+} is shown to bind specifically to ATPase subunit ATP5B, and the silencing of this subunit induces similar effects on AMPK activation, AMP/ATP ratio, and restoration of hyperglycemia-induced mitochondrial fragmentation. Finally, the authors show that Cr^{3+} feeding is able to protect against hyperglycemia stress in diabetic mice.

The manuscript is technically and conceptually sound. However, a few questions remain to better understand the mechanism proposed.

[General Response to Reviewer #4]

We truly appreciate the reviewer's favorable comments and helpful suggestions on the present study as well as recognition on the major merits of our work. These suggestions are immensely useful for us to improve the manuscript.

1- Cr^{3+} dependent inhibition of the mitochondrial ATPase is expected to alter two key parameters such as the mitochondrial membrane potential and the generation of superoxide. The contribution of these parameters needs to be measured and discussed.

[Response 1] We truly thank the reviewer's helpful comments. We measured the mitochondrial membrane potential and mitochondrial ROS in HepG2 cells with and without treatment of CrCl_3 as suggested. As shown in **Supplementary Fig. 10** (in the revised version), hyperglycaemia increases the mitochondrial ROS levels of HepG2 cells to 1.5 folds. The treatment of 50 μM CrCl_3 could inhibit the hyperglycaemia-induced mitochondrial ROS by 53.9%, and with the increasing of CrCl_3 concentration, such an inhibition of became more obvious with 100 μM CrCl_3 nearly completely normalized the elevated mitochondrial ROS. While for the mitochondrial membrane potential (MMP), hyperglycaemia treatment leads to decreased MMP to 55.1% (the MMP under normal glucose condition is set as 100%) and the supplementation of CrCl_3 could reverse the dropped MMP back to normal level (**Supplementary Fig. 10b**). These new data and discussion have been incorporated into the revised manuscript.

Supplementary Fig. 10 Effect of CrCl_3 treatment on mitochondrial ROS and mitochondrial membrane potential (MMP) in HepG2 cells under high glucose (HG). (a) Mitochondrial ROS (n = 3). (b) Mitochondrial membrane potential (n = 3). Normal glucose (NG) is used as control. Results are shown

as mean \pm SEM. The asterisks denote significant difference of different groups (* $P < 0.05$, ** $P < 0.01$, *** $P < 0.001$ and **** $P < 0.0001$). NS, not significant ($P > 0.05$).

2- It is intriguing both that hyperglycemia induces mitochondrial fragmentation and that Cr^{3+} restores it. The mechanism by which Cr^{3+} -binding to ATP5B impacts mitochondrial dynamics needs to be further discussed.

[Response 2] We agree with the reviewer that further discussion is needed. As suggested, we have added on paragraph to further discuss the mechanism by which Cr^{3+} -binding to ATP5B affects mitochondrial dynamics (page 6, line 59-62 and page 7, line 1-8).

3- Since the mitochondrial ATPase plays an important role in cristae formation/stabilization, it would be important to test whether Cr^{3+} -binding to ATP5B prevents the required ATPase oligomerization involved in the cristae formation process.

[Response 3] We truly thank the reviewer's insightful suggestion. The fully assembled F-type ATP synthase has been known to exist as dimers on the inner mitochondrial membrane, and this dimerization is known to influence the characteristic cristae formation by the membrane (*EMBO J.* 2002, 21, 221; *PNAS*, 2012, 109, 13602). As suggested, we extracted the ATP synthase by digitonin from HeLa cell mitochondria and examined the effect of Cr(III)-binding on the oligomeric states of ATP synthase by blue native blue page.

As shown in **Supplementary Fig. 9** in the revised version, after resolving the samples on a $4\% \pm 16\%$ gradient BNP gel capable of resolving proteins at a molecular weight range between 20 kDa to approximately 1,200 kDa (**Supplementary Fig. 9, lane 1**). Silver staining revealed prominent bands at the sizes corresponding to dimeric ($\sim 1,000$ kDa) and monomeric (~ 600 kDa) forms of ATP synthase. After treatment of increasing amount of Cr^{3+} , no significant change of band intensity of either dimer or monomer form ATP synthase, indicating that the oligomerization state of the enzyme was not affected by Cr^{3+} binding. The new data (**Supplementary Fig. 9**) and discussion have been incorporated into the revised manuscript.

Supplementary Fig. 9 Effect of Cr^{3+} treatment on ATP synthase oligomerization state. Digitonin protein extracts of mitochondria isolated from HeLa cells were separated by blue native page (BNP).

REVIEWER COMMENTS

Reviewer #1 (Remarks to the Author):

Since the initial submission of the manuscript the authors have used the time effectively to collect additional data in support of the metabolic effects of chromium(III). This effort is laudable.

The question whether the probe itself is responsible for mitochondrial targeting remains unanswered and so remains the coordination chemistry of chromium(III) in the probe.

If the authors cannot answer these questions, minimally they need to bring the underlying implications to the attention of the readers. Octahedral chromium(III) complexes are kinetically rather inert. The reported ligand exchange rates amount to a $t_{1/2}$ of about 400 hours. Thus, the question whether the geometry of the fluorescent probe has a specific coordination environment that enables faster exchange seems highly relevant. A discussion of the differences between CrCl_3 and the Cr complex of the probe is also important in terms of the concentrations required in the investigations. Cells are rather impermeant to Cr(III), and Cr(III) is widely used for labelling the cell membrane. There should also be a comment on characterized Cr(III) ATP complexes and their properties. Is Cr(III)ATP directly involved in the inhibition mechanism?

A major issue to address in the message of the article is to distinguish physiological effects from pharmacological effects by instructing the reader about how low blood and tissue concentrations of chromium usually are, what the recommended daily intake is versus the intake from chromium supplements that can have chromium amounts in the range of 200 to 1000 micro-g. Such a discussion needs to connect to the effects observed: the measured inhibition constant for the ATPase is rather weak with $K > 200$ microM (Figure 2a). It is not high affinity. It would require rather high local chromium concentrations. For Figure 2b: Which chromium concentration was used? If $t_{1/2}$ is indeed 400 hours (see above) then 12 hours are not enough!

I misspelled the oxidation state in my review as Cr(IV) instead of Cr(VI). Unfortunately, the authors were uncritical in not noticing the misspelling. The article needs to discuss briefly the toxicity of Cr(VI) in the form of chromate including a critical comment on the postulated formation from chromium(III) in the cell and some references. The authors need to change the terminology: Chromate is a toxicant and not a toxin. Mentioning toxicology is very important because industry could easily see the data provided in this manuscript as proof of principle for efficacy and safety of chromium supplements.

Reviewer #3 (Remarks to the Author):

Reviewer's comments on the revised manuscript by Wang et al. "Mitochondrial ATP synthase as the first direct molecular target of 3 chromium(III) to ameliorate hyperglycaemia stress"

The authors have answered my questions and indeed resolved some doubts I had about the study in itself. These initially revolved around the specificity of their assay, and I would like to thank the authors that they replied to this, although my assessment primarily concerned their mass-spectrometry data.

I appreciate that the authors have added sentences to the Introduction section to the effect that “Given that NTA-AC itself is not cell permeable and only metal bound Cr-NTA-AC (namely Mn⁺-NTA-AC or Mn⁺-TRACER) could enter cells to label proteins, and different metal binding proteins have been identified previously when different metal ions were bound to NTA-AC, which suggests the bindings of the probe to proteins are metal orientated.”

I also appreciate that the authors showed that GAPDH does not respond to Cr³⁺.

All in all, the results indeed point towards that ATB5B binds/interact with Cr³⁺.

I would nonetheless like to ask the authors whether they really think that it is unimportant – particularly in the light of the current debate regarding Cr³⁺ and its possible function in nutrition – to show that the measured effects on ATP5B are caused specifically by Cr³⁺ and that no other transition metals induce a similar effect. It would be highly beneficial for the manuscript if the authors could show that no other (transition) metal interacts with e.g. ATB5B (or, if it does so then with a higher K_d than Cr³⁺).

Reviewer #4 (Remarks to the Author):

The authors have responded to my previous criticism, and I believe that also to most comments by the other reviewers. Therefore, I support the acceptance of this revised manuscript.

Reviewer #5 (Remarks to the Author):

I have assessed the authors response to the reviewer's comments and believe they have adequately addressed the major concerns in relationship to determining the mechanism by which Cr³⁺ activates AMPK.

Re: “Mitochondrial ATP synthase as the first direct molecular target of chromium(III) to ameliorate hyperglycaemia stress” (NCOMMS-21-25340A)

REVIEWER COMMENTS

Reviewer #1 (Remarks to the Author):

Since the initial submission of the manuscript the authors have used the time effectively to collect additional data in support of the metabolic effects of chromium(III). This effort is laudable.

[General Response to Reviewer #1]

We truly appreciate the reviewer’s favorable comments and recognition of our efforts.

The question whether the probe itself is responsible for mitochondrial targeting remains unanswered and so remains the coordination chemistry of chromium(III) in the probe.

If the authors cannot answer these questions, minimally they need to bring the underlying implications to the attention of the readers. Octahedral chromium(III) complexes are kinetically rather inert. The reported ligand exchange rates amount to a $t_{1/2}$ of about 400 hours. Thus, the question whether the geometry of the fluorescent probe has a specific coordination environment that enables faster exchange seems highly relevant. A discussion of the differences between CrCl_3 and the Cr complex of the probe is also important in terms of the concentrations required in the investigations. Cells are rather impermeant to Cr(III), and Cr(III) is widely used for labelling the cell membrane. There should also be a comment on characterized Cr(III) ATP complexes and their properties. Is Cr(III)ATP directly involved in the inhibition mechanism?

[Response]

We truly appreciate the reviewer’s helpful suggestions. These suggestions are immensely useful for us to improve the manuscript.

For the question whether the probe itself is responsible for mitochondrial targeting remains unanswered:

We first demonstrated that the blue fluorescence signals of Cr^{3+} -NTA-AC were co-localized with the red fluorescence signals of MitoTracker (Fig. 1b), implying that the Cr^{3+} -associated proteins are mainly localized in the mitochondria of HepG2 cells. We further verified this by measuring chromium distribution with ICP-MS (Supplementary Fig. 3), which showed that treatment of HepG2 cells with CrCl_3 resulted in the accumulation of chromium mainly in mitochondrial, in line with our fluorescence imaging data. Equally important, we showed that CrCl_3 could rescue high glucose induced mitochondrial fission or fusion of HepG2 cells (Fig. 3), while deletion of ATP5B abolished such effects. Collectively, we are confident to conclude that Cr^{3+} targeting mitochondrial is a general phenomenon regardless of using the probe or chromium compounds.

For the coordination chemistry of chromium(III) in the probe:

Based on our previous published crystal structure of a Fe-probe (Fe^{3+} -NTA-AC) bound to transferrin (*Metallomics* 2018, 10,77-82), the probe bound to the targeted protein as a whole molecule with one carboxylate group from NTA being dissociated from Fe. Since Cr(III) has similar coordination behavior to Fe(III), it is reasonable to assume that the Cr^{3+} -NTA-AC fluorescent probe bound to the proteins via two or three vacant sites (one carboxylate group from NTA might dissociated as well) with the confined geometry and coordination environment, which may facilitate fast ligand exchange with the biomolecules inside the cells, enabling higher cell permeability and rapid labelling of proteins. The higher bioavailability of Cr^{3+} -NTA-AC fluorescent probe explains why much lower concentration of Cr^{3+} -NTA-AC is needed for cell study compared to that of CrCl_3 . Moreover, the reactivity of Cr(III) complex is very different *in vitro* and inside cells. Additional discussion was incorporated into the

revised manuscript (page 3, line 12-13).

We agree with the reviewer that octahedral Cr(III) complexes are kinetically rather inert in vitro (e.g., the rate of water exchange on Cr(III) is $2.6 \times 10^{-6} \text{ s}^{-1}$ at 298 K). The half-life of Cr(III) in solution is 7.2 to 43.2 hrs (“Reviews of the Environmental Effects of Pollutants: Chromium (III)”, 2013, page 27). The half-lives of Cr(III) compounds range from 0.5 to 12 hrs in blood (fast compartment), 1 to 14 days in storage organs like the liver and spleen (medium), and 3 to 12 months in other solid tissues (slow compartment) (*Am. J. Physiol. Regul. Integr. Comp. Physiol.*, 1983, 244, R445-R454).

We also agree with the reviewer that cells are rather impermeant to Cr(III). Thus, CrCl_3 was poorly absorbed (estimated 0.13% bioavailability) and rapidly eliminated in urine (excretion half-life, approximately 10 hr) (*Toxicol. Appl. Pharmacol.*, 1996, 141, 145-158). The low bioavailability of Cr(III) is a very important issue to be addressed to further improve the pharmacological effects of Cr(III) supplements.

Based on previous reports, Cr(III)ATP compounds are directly involved in the inhibition mechanism. In detail, β, γ -Cr(III)ATP and α, β, γ -Cr(III)ATP are competitive inhibitors of ATP synthase with respect to Mg(II)ATP (*Arch. Biochem. Biophys.*, 1987, 258, 482-490; *Microbiol. Mol. Biol. Rev.*, 2008, 72, 590-641). β, γ -Cr(III)ATP and α, β, γ -Cr(III)ATP inhibit F1 by binding at the catalytic site and α, β -Cr(III)ATP by binding at a regulatory site. The binding sites show no significant selectivity for the steric arrangement of the Cr(III) complexes. β, γ -Cr(III)ATP and α, β, γ -Cr(III)ATP bind to the catalytic site with the same affinity, although they have different steric arrangements of the chromium, i.e., β, γ -Cr(III)ATP with monocyclic coordination at the metal ion and α, β, γ -Cr(III)ATP with bicyclic coordination. A description of characterized Cr(III) ATP complexes and their properties is added (page 6, line 50-56).

A major issue to address in the message of the article is to distinguish physiological effects from pharmacological effects by instructing the reader about how low blood and tissue concentrations of chromium usually are, what the recommended daily intake is versus the intake from chromium supplements that can have chromium amounts in the range of 200 to 1000 micro-g. Such a discussion needs to connect to the effects observed: the measured inhibition constant for the ATPase is rather weak with $K_i > 200 \text{ microM}$ (Figure 2a). It is not high affinity. It would require rather high local chromium concentrations. For Figure 2b: Which chromium concentration was used? If $t_{1/2}$ is indeed 400 hours (see above) then 12 hours are not enough !

[Response]

We truly appreciate the reviewer's suggestion.

The concentrations of chromium are normally in the range of 0.11–16.4 ng/mL and 3-800 ng/g in human blood and different tissues respectively in non-exposed populations (*Biol. Trace Elem. Res.*, 2018, 186, 370-378).

For the recommended daily intake, adequate intake (AI) was set as an estimated safe and adequate daily dietary intake for chromium. The intake for an adult (19-50 years) is 35 μg and 25 μg daily for men and women respectively. Men and women older than 50 years require slightly less, at 20 and 30 μg daily respectively (*Biol. Trace Elem. Res.*, 2018, 186, 370-378).

To select the concentration of Cr(III) for cell study, we first examined the cytotoxicity of Cr(III) against HepG2 cells, our results showed that negligible cytotoxicity of CrCl_3 at the concentration lower than 1000 μM was observed (Supplementary Fig. 3a). We therefore did the cell studies with Cr(III) in a dose-dependent manner in the range of 50-200 μM . Our ICP-MS result showed that the intracellular concentration of Cr(III) in HepG2 cells after treatment of Cr(III) is at around 100 $\mu\text{g/L}$, which is similar to the serum concentration (hundreds $\mu\text{g/L}$) and tissue concentration (hundreds ng/g) of Cr(III) in mice exposed to Cr(III) diet (*Biol. Trace Elem. Res.*, 2019, 187, 192-201). Moreover, we also observed a ca. 25% decrease in the activity of ATP synthase in Cr^{3+} -treated *db/db* mice liver (Fig. 4g), which is again very similar to the cell study results.

For Fig. 2b, we have used 200 μM of Cr(III). In our study, we examined the time-dependent uptake of Cr(III) in HepG2 cells by ICP-MS. The result showed that the intracellular concentrations of Cr(III) increased gradually within the first 8 hrs of treatment and then levelled off until 20 hrs (Supplementary Fig. 3b). We therefore chose the treatment time from 1 to 12 hrs in the enzymatic activity test based on the uptake kinetics rather than the half time ($t_{1/2}$).

Additional discussions on these parts are added in the revised manuscript (page 2, line 41-42; page 6, line 10-11).

Supplementary Fig. 3a Cytotoxicity test of CrCl₃ against HepG2 cells. Negligible cytotoxicity of CrCl₃ at concentration up to 1000 μM was noted (n=3).

I misspelled the oxidation state in my review as Cr(IV) instead of Cr(VI). Unfortunately, the authors were uncritical in not noticing the misspelling. The article needs to discuss briefly the toxicity of Cr(VI) in the form of chromate including a critical comment on the postulated formation from chromium(III) in the cell and some references. The authors need to change the terminology: Chromate is a toxicant and not a toxin. Mentioning toxicology is very important because industry could easily see the data provided in this manuscript as proof of principle for efficacy and safety of chromium supplements. While the trivalent Cr(III) is only toxic at high concentrations, hexavalent Cr(VI), a strong oxidizer, is considered toxic to humans and the environment at $\mu\text{g}/\text{L}$ concentrations

[Response]

We truly agree with the reviewer that mentioning the toxicology is very important.

We have revised the oxidation state of Cr(IV) to Cr(VI) (page 6, line 2).

A discussion about the toxicity of Cr(VI) and a critical comment on the postulated formation of Cr(III) in the cell was added (page 6, line 3-8).

We have changed the terminology: Chromate is a toxicant and not a toxin, in the revised manuscript (page 6, line 4).

Reviewer #3 (Remarks to the Author):

Reviewer's comments on the revised manuscript by Wang et al. "Mitochondrial ATP synthase as the first direct molecular target of 3 chromium(III) to ameliorate hyperglycaemia stress"

The authors have answered my questions and indeed resolved some doubts I had about the study in itself. These initially revolved around the specificity of their assay, and I would like to thank the authors that they replied to this, although my assessment primarily concerned their mass-spectrometry data.

I appreciate that the authors have added sentences to the Introduction section to the effect that "Given that NTA-AC itself is not cell permeable and only metal bound Cr-NTA-AC (namely Mn⁺-NTA-AC or Mn⁺-TRACER) could enter cells to label proteins, and different metal binding proteins have been

identified previously when different metal ions were bound to NTA-AC, which suggests the bindings of the probe to proteins are metal orientated.”

I also appreciate that the authors showed that GAPDH does not respond to Cr³⁺. All in all, the results indeed point towards that ATB5B binds/interact with Cr³⁺.

[General Response to Reviewer #3]

We truly appreciate the reviewer's favorable comments.

I would nonetheless like to ask the authors whether they really think that it is unimportant – particularly in the light of the current debate regarding Cr³⁺ and its possible function in nutrition – to show that the measured effects on ATP5B are caused specifically by Cr³⁺ and that no other transition metals induce a similar effect. It would be highly beneficial for the manuscript if the authors could show that no other (transition) metal interacts with e.g. ATB5B (or, if it does so then with a higher K_d than Cr³⁺).

[Response]

We truly appreciate the reviewer's constructive suggestion.

As this study aims to unveil the molecular targets and pharmacological effects of exogenous Cr(III) ions. We measured the activity of ATP synthase (under hyperglycaemia stress) in cells supplemented with gradient amounts of Cr³⁺, instead of *in vitro*, and we did observe a dose- and time-dependent inhibition of ATP synthase activity by Cr³⁺. At 200 μM Cr³⁺, about 37% activity was inhibited compared to that without addition of Cr³⁺ (Fig. 2a and 2b). This is sufficient to demonstrate the observed effect is attributable to Cr³⁺, but not other metal ions.

ATP synthase is a multi-subunit, membrane-associated protein complex that catalyzes the phosphorylation of ADP to ATP at the expense of a proton motive force and with the aid of Mg²⁺ ion in the catalytic center of ATP5B. As reported previously, other transition metal ions, such as Zn²⁺, Co²⁺, Mn²⁺ and Ca²⁺ ions, could inhibit ATP synthase in a competitive manner *in vitro* (*J. Biol. Chem.*, 1992, 267, 10252-10257; *Microbiol. Mol. Biol. Rev.*, 2008, 72, 590-641.), indicating that these transition metal ions could bind to the Mg²⁺ sites and inhibit ATP5B, which is similar to the effect of Cr³⁺.

Based on the hard and soft (Lewis) acids and bases theory (HSAB), hard acids strongly favor hard bases. The metal binding site of ATP5B is composed of -OH from threonine, -COO⁻ from glutamic acid and aspartic acid, and PO₄³⁻ from ATP/ADP. All these ligands are hard bases. Mg²⁺ (and Ca²⁺) and Cr³⁺ are hard acids, and other transition metals are intermediate acid (such as Co²⁺, Mn²⁺ and Zn²⁺ ions). Based on the HSAB concept, both hard acids and intermediate acids could interact with hard bases. Thus, binding of other transition metal ions to the active site of ATP5B in a similar manner with Cr³⁺ and consequently inhibit its activity is not unexpected.

However, in the current study, we focus on the molecular targets and pharmacological effects of exogenous Cr(III) ions, the observed *in vitro* binding of other transition metal ions may not occur in cells owing to highly regulated transition metal ions intracellularly (*i.e.*, very low concentrations of free metal ions in its low molecular species). Nevertheless, we quote previous studies on the inhibition of ATP synthase by these transition metal ions *in vitro* in the revised manuscript (page 6, line 54-56).

Reviewer #4 (Remarks to the Author):

The authors have responded to my previous criticism, and I believe that also to most comments by the other reviewers. Therefore, I support the acceptance of this revised manuscript.

[General Response to Reviewer #4]

We truly appreciate the reviewer's favorable comments.

Reviewer #5 (Remarks to the Author):

I have assessed the authors response to the reviewer's comments and believe they have adequately addressed the major concerns in relationship to determining the mechanism by which Cr³⁺ activates AMPK.

[General Response to Reviewer #5]

We truly appreciate the reviewer's favorable comments.

REVIEWER COMMENTS

Reviewer #1 (Remarks to the Author):

The authors use chemical biology and metalloproteomics approaches to the hitherto unresolved issue of the cellular targets of chromium. These approaches are novel with a significant number of chemical biological and biological experiments in cells and mice. With their finding of mitochondrial targets of chromium, they advance the field, which has not seen much progress in this regard, significantly.

The questions raised in the previous round of submission, I believe, have been answered.

The authors should distinguish charge and valence state. Instead of Cr³⁺ they need to write Cr(III) at many but not all instances in the manuscript.

First sentence:among the other physiologically important transition metals, not just transition metal ions as there are others.

Chromodulin is not a protein, it is a peptide.

Editing of grammar (omission of article, singular vs plural in verb etc) is essential. Occasionally, some improvement of language also would be advisable.

Reviewer #3 (Remarks to the Author):

Reviewer's comments on the revised manuscript by Wang et al. "Mitochondrial ATP synthase as the first direct molecular target of chromium(III) to ameliorate hyperglycaemia stress" (NCOMMS-21-25340B)

The authors have addressed the point that I made on their first revision. However, I would have liked to see the experiments that I suggested. In the present revised version the authors have added a sentence and refer to the literature where it is stated that the ATP synthase is also inhibited by other transition metals. I have to state clearly that – despite the fact the authors and the laboratory are well regarded and are internationally visible in respect of the studies presented here to identify metal-binding proteins in cells – I have serious doubts that this study identifies a natural binding partner of Cr³⁺. This is because, to the best of my knowledge, no protein or other biomolecule in human cells binds chromium or requires it for activity, stability, interaction, etc. In this study, I do not see any convincing results to indicate that the rôle of chromium (in the food supply) cannot be played by other transition metals.

Re: “Mitochondrial ATP synthase as the first direct molecular target of chromium(III) to ameliorate hyperglycaemia stress” (NCOMMS-21-25340B)

REVIEWER COMMENTS

Reviewer #1 (Remarks to the Author):

The authors use chemical biology and metalloproteomics approaches to the hitherto unresolved issue of the cellular targets of chromium. These approaches are novel with a significant number of chemical biological and biological experiments in cells and mice. With their finding of mitochondrial targets of chromium, they advance the field, which has not seen much progress in this regard, significantly.

The questions raised in the previous round of submission, I believe, have been answered.

[General Response to Reviewer #1]

We truly appreciate the reviewer’s favorable comments and recognition of our efforts.

The authors should distinguish charge and valence state. Instead of Cr³⁺ they need to write Cr(III) at many but not all instances in the manuscript.

First sentence: ...among the other physiologically important transition metals, not just transition metal ions as there are others.

Chromodulin is not a protein, it is a peptide.

Editing of grammar (omission of article, singular vs plural in verb etc) is essential. Occasionally, some improvement of language also would be advisable.

[Response]

We truly appreciate the reviewer’s helpful suggestions. These suggestions are immensely helpful for us to improve the manuscript.

- We have distinguished charge and valence state of chromium and revised Cr³⁺ to Cr(III) when meaning valence state.
- We have revised the first sentence as suggested.
- We have revised “chromium-binding protein (chromodulin)” to “chromium-binding peptide (chromodulin)”.
- We have improved the grammar extensively as suggested.

Reviewer #3 (Remarks to the Author):

Reviewer's comments on the revised manuscript by Wang et al. “Mitochondrial ATP synthase as the first direct molecular target of chromium(III) to ameliorate hyperglycaemia stress” (NCOMMS-21-25340B)

The authors have addressed the point that I made on their first revision. However, I would have liked to see the experiments that I suggested. In the present revised version the authors have added a sentence and refer to the literature where it is stated that the ATP synthase is also inhibited by other transition metals. I have to state clearly that – despite the fact the authors and the laboratory are well regarded and are internationally visible in respect of the studies presented here to identify metal-binding proteins in cells – I have serious doubts that this study identifies a natural binding partner of Cr³⁺. This is because, to the best of my knowledge, no protein or other biomolecule in human cells binds chromium or requires it for activity, stability, interaction, etc. In this study, I do not see any convincing results to indicate that the rôle of chromium (in the food supply) cannot be played by other transition metals.

[Response]

We truly appreciate the reviewer’s comments and kind suggestion.

As suggested by the reviewer, we measured the bindings of other essential transition metal ions to ATP5B, including Zn^{2+} , Mn^{2+} , Cu^{2+} , Co^{2+} , Ni^{2+} , and Fe^{3+} , under identical condition to Cr^{3+} . As shown in Supplementary Fig. 16a, Mn^{2+} , Zn^{2+} , and Cu^{2+} bind to ATP5B in a dose-dependent manner but the binding is much weaker than Cr^{3+} . For Co^{2+} , Ni^{2+} and Fe^{3+} , negligible binding to ATP5B is noticed even in the presence of 20 eq. of metal ions. We further performed the competition study by adding 10 eq. different metal ions into the Cr-bound ATP5B. As shown in Supplementary Fig. 16b, only Mn^{2+} , Zn^{2+} , and Cu^{2+} could partially replace Cr^{3+} from Cr-ATP5B, but not the other metal ions. Taken together, the bindings of these transition metal ions to ATP5B are not as strong as Cr^{3+} as evidenced by the binding affinity and competition studies. Thus, they are unlikely to compete with Cr^{3+} towards binding to ATP synthase, particularly in cells owing to low cellular concentrations of free transition metal ions (i.e., tightly regulated).

Supplementary Fig. 16 | Binding capability of different transition metal ions to ATP5B. (a) Dose-dependent binding of different metal ions to ATP5B (n= 3). (b) The metal contents in Cr-ATP5B upon supplementation of different metal ions (n= 3). Results are shown as mean \pm SEM.

REVIEWERS' COMMENTS

Reviewer #3 (Remarks to the Author):

Reviewer's comments on the third revised version of the manuscript by Wang et al. "Mitochondrial ATP synthase as the first direct molecular target of 3 chromium(III) to ameliorate hyperglycaemia stress"

I appreciate the author's efforts in order to demonstrate that the binding affinity of Cr³⁺ towards the ATPase is stronger than other transition metals and that these results are part of the revised manuscript.

Based on the results I am happy to state that the authors have dispelled my doubts and therefore I therefore, I recommend the publication of the study in Nature Communications.